# Generalization Bound of Gradient Descent for Non-Convex Metric Learning

**Mingzhi Dong**[1,2,3*]    **Xiaochen Yang**[3*]    **Rui Zhu**[4]    **Yujiang Wang**[5]    **Jing-Hao Xue**[3]

[1]School of Computer Science, Fudan University, Shanghai, China
[2]Shanghai Key Laboratory of Data Science, Fudan University, Shanghai, China
[3]Department of Statistical Science, University College London, London, UK
[4]The Business School (formerly Cass), City, University of London, London, UK
[5]Department of Computing, Imperial College London, London, UK

mingzhidong@gmail.com    {xiaochen.yang.16, jinghao.xue}@ucl.ac.uk
rui.zhu@city.ac.uk    yujiang.wang14@imperial.ac.uk

## Abstract

Metric learning aims to learn a distance measure that can benefit distance-based methods such as the nearest neighbor (NN) classifier. While considerable efforts have been made to improve its empirical performance and analyze its generalization ability by focusing on the data structure and model complexity, an unresolved question is how choices of algorithmic parameters, such as the number of training iterations, affect metric learning as it is typically formulated as an optimization problem and nowadays more often as a non-convex problem. In this paper, we theoretically address this question and prove the agnostic Probably Approximately Correct (PAC) learnability for metric learning algorithms with non-convex objective functions optimized via gradient descent (GD); in particular, our theoretical guarantee takes the iteration number into account. We first show that the generalization PAC bound is a sufficient condition for agnostic PAC learnability and this bound can be obtained by ensuring the uniform convergence on a densely concentrated subset of the parameter space. We then show that, for classifiers optimized via GD, their generalizability can be guaranteed if the classifier and loss function are both Lipschitz smooth, and further improved by using fewer iterations. To illustrate and exploit the theoretical findings, we finally propose a novel metric learning method called *S*mooth *M*etric and representative *I*nstance *LE*arning (SMILE), designed to satisfy the Lipschitz smoothness property and learned via GD with an early stopping mechanism for better discriminability and less computational cost of NN.

## 1   Introduction

A good measure of distance between instances is important to many machine learning algorithms, such as the nearest neighbor (NN) classifier and $k$-means clustering. As it is difficult to handcraft an optimal distance for each task, metric learning appears as an appealing technique to learn the distance metric automatically and directly from the data. The most widely studied metric is the Mahalanobis distance and it is often learned via an optimization problem [51, 17, 49]. To enhance the discriminability of the learned metric, various loss functions have been designed, considering the local property of heterogeneous data [15, 47, 23, 5, 38, 54, 44, 12] and the nonlinear geometry of the sample space [24, 58, 8]. Meanwhile, to achieve good generalization and robustness, different regularizations have been imposed to control the model complexity [31, 26, 50, and references therein]. In addition to methodological advances, theoretical guarantees of metric learning algorithms, as well as guarantees

---

[*]The first two authors contributed equally to this work.

of metric-based classifiers [3, 19], have been provided. In particular, generalization bounds have been founded on the complexity measure of the model class [55, 4, 7, 46, 33, 53], algorithmic stability [27, 19, 16], and algorithmic robustness [2]. The intrinsic complexity of the dataset has also been considered in recent studies [46, 33].

While the data structure and model complexity play a vital role in metric learning, an equally important but as yet poorly understood factor is the choice of optimization algorithms and the associated parameters [42]. For example, when metric learning is formulated as a non-convex problem and optimized by using the gradient descent algorithm, its solution is inevitably influenced by factors such as the learning rate and the number of training iterations; the optimal models with respect to different optimization parameters will then exhibit different generalization behavior.

Therefore, the goal of this paper is to provide a new route to theoretical exploration and exploitation of the effect of the gradient descent (GD) algorithm on metric learning methods. To this end, we establish a generalization bound which suggests that early stopping, smooth classifier and smooth loss function have crucial influence on the generalization error. We highlight that the proposed theoretical techniques do not take advantage of any property of convex optimization and are not specific to metric learning methods; they can be used to study the generalization ability of classification algorithms with non-convex objectives. The contributions of this paper are fourfold.

1. We show that the generalization Probably Approximately Correct (PAC) bound is a sufficient condition for a parametric hypothesis class to be agnostic PAC learnable (Theorem 1). Compared with the widely studied uniform convergence condition, the generalization PAC bound is a weaker notion but has the capability to analyze the influence of algorithmic parameters.

2. To facilitate the derivation of the generalization PAC bound of a hypothesis class, we propose a new decomposition theorem to decompose the bound into two terms that can be easily guaranteed (Theorem 2). The first term constrains the space of the estimated parameters of the hypothesis, reducing it from the entire parameter space to a high-confidence subset of the parameter space. The second term considers the uniform convergence condition of the concentrated subset.

3. Based on the decomposition theorem, we obtain the generalization PAC bound for classifiers learned with the gradient descent algorithm (Theorem 3). The bound shows that the generalization gap increases over iterations, thus providing a theoretical support for the practical use of early stopping. Moreover, it shows that a Lipschitz smooth (i.e. Lipschitz continuous of the gradient) classifier and a Lipschitz smooth loss function are sufficient to guarantee good generalization.

4. We propose a novel metric learning method as a concrete example of using the generalization PAC bound. When classifying a test instance, the NN classifier has to store the entire training set and calculate its distances to all training instances, thereby incurring high storage and computational costs. To reduce these costs and improve the generalization performance, we propose to simultaneously learn the distance metric and few representative instances which serve as the reference points for testing; the new method is called *Smooth Metric and representative Instance LEearning* (SMILE). More specific, to ensure good test performance, SMILE adopts a Lipschitz smooth classifier and loss function and is optimized via GD with a designed early stopping mechanism. The method is evaluated on 12 datasets and shows competitive performance against existing methods.

## 1.1 Related work

**Generalization bound of GD with early stopping**   Early stopping in regularizing the model complexity and its effect on the generalization ability have been extensively studied for a wide range of methods, such as perceptron algorithm [9], kernel regression [52], and deep neural networks [36]. Our algorithm-dependent PAC bound is motivated by [21], which proves the generalizability for models learned with stochastic GD. Founded on uniform stability, [21] can only bound the generalization gap in expectation. In contrast, by using the generalization PAC bound, we are able to construct a high probability bound, i.e. the agnostic PAC learnability.

**Generalization bounds for metric learning algorithms**   The majority of generalization guarantees for metric learning algorithms rely on the uniform convergence property of the hypothesis class, e.g. [19, 7]. In this paper, by viewing the learned hypothesis as a random variable, we only require the convergence property to hold with high probability. This condition is easier to be satisfied by

ruling out the hypotheses that will not be explored by an optimization algorithm, and is useful for analyzing the effect of early stopping and other operators that limit search in the hypothesis class.

**Generalization bounds for Lipschitz classifiers and losses** [32, 18] use Lipschitz functions as large margin classifiers in general metric spaces and provide generalization bounds for Lipschitz classifiers. Our theoretical guarantee is different from their work in two aspects. First, in [32, 18] the input space of the Lipschitz constant is the data space, whereas in our paper it is the parameter space. Second, owing to this difference, the generalization bound obtained in our work has a faster convergence in most cases. [46] derives the generalization bound for metric learning algorithms with Lipschitz continuous loss functions. However, when taking the influence of GD into account, Lipschitz smoothness is also important to be considered for generalization purposes. [53] uses a smooth loss function to obtain a fast generalization. However, their work requires the objective function to be strongly convex, which is different from our focus on non-convex problems.

**Metric learning with representative instances** Reducing the amount of necessary training data as a way of reducing the storage and computational costs of NN has been extensively studied, e.g. in [35, 57, 40]. Among these methods, DMPL [28], SNC [29] and ProtoNN [20] are most relevant to our work, as they also learn the distance metric and representative instances simultaneously. Our method differs from them in the loss function and regularization terms, both of which are designed in our work to provide a theoretical guarantee on the classification performance.

## 2 Preliminaries

### 2.1 Notation

This paper focuses on binary classification problems. Let $\boldsymbol{z}^n = \{\boldsymbol{z}_i = (\boldsymbol{x}_i, y_i), i = 1, \ldots, n\} \in \mathcal{Z}^n$ denote the set of $n$ independent and identically distributed (i.i.d.) training instance and label pairs, sampled from an unknown joint distribution $p(\boldsymbol{z}) = p(\boldsymbol{x}, y)$. Let $h(\boldsymbol{x}, \boldsymbol{w})$ be a function with instance $\boldsymbol{x}$ and parameter $\boldsymbol{w} \in \mathcal{W} \subseteq \mathbb{R}^Q$. The output of $h(\boldsymbol{x}, \boldsymbol{w})$ is restricted to be a real value; $\text{sign}\,[h(\boldsymbol{x}, \boldsymbol{w})]$ returns the classification decision, where $\text{sign}[\cdot]$ denotes the sign function.

During the training process of the classifier, given $\boldsymbol{z}^n$, a *classifier* or *hypothesis* $\hat{h}$ can be obtained from an optimization algorithm, such as GD. With a parametric classifier, $\hat{h}$ can be fully represented by $\hat{\boldsymbol{w}}$ as $\hat{h} = h(\boldsymbol{x}, \hat{\boldsymbol{w}})$. $R_n(\boldsymbol{z}^n, \hat{h}) := R_n(\boldsymbol{z}^n, \hat{\boldsymbol{w}}) := \frac{1}{n}\sum_i r(\boldsymbol{z}_i, \hat{\boldsymbol{w}}) := \frac{1}{n}\sum_i l(h(\boldsymbol{x}_i, \hat{\boldsymbol{w}}), y_i)$ is called the *training error*, where $r(\cdot, \cdot)$ denotes the risk function, $l(\cdot, \cdot)$ denotes the loss function, and they are assumed to be non-negative. Let $\boldsymbol{s} \in \mathcal{S}$ denote a fixed setting of the algorithm, including e.g. the initial values, the number of iterations and the learning rate. The relationship between $\hat{\boldsymbol{w}}$ and $\boldsymbol{z}^n$ is represented as $\hat{\boldsymbol{w}} = \boldsymbol{m}(\boldsymbol{z}^n; \boldsymbol{s})$, where $\boldsymbol{m} : \mathcal{Z}^n \times \mathcal{S} \to \mathcal{W}$; $\boldsymbol{m}(\boldsymbol{z}^n; \boldsymbol{s})$ will sometimes be abbreviated to $\boldsymbol{m}(\boldsymbol{z}^n)$ for notational simplicity. Since $\hat{\boldsymbol{w}}$ is a function of random samples $\boldsymbol{z}^n$, it is a random variable.

During the test process, a test pair $\boldsymbol{z}' = (\boldsymbol{x}', y')$ is sampled from the same unknown distribution $p(\boldsymbol{z})$. The predicted value $h(\boldsymbol{x}', \hat{\boldsymbol{w}})$ will be compared with the true label $y'$ to evaluate the performance of the algorithm. $R(\hat{h}) := R(\hat{\boldsymbol{w}}) := \mathbb{E}_{\boldsymbol{z}'} r(\boldsymbol{z}', \hat{\boldsymbol{w}}) := \mathbb{E}_{\boldsymbol{z}'} l(h(\boldsymbol{x}', \hat{\boldsymbol{w}}), y')$ is called the *test error*.

The gap between the training error and the test error, $R(\hat{\boldsymbol{w}}) - R_n(\boldsymbol{z}^n, \hat{\boldsymbol{w}})$, is called the *generalization gap*. A good classifier should have small training error and small generalization gap so as to perform well on test instances.

Let $\|\boldsymbol{a}\|_2$ denote the $L_2$-norm of a vector $\boldsymbol{a}$ and $\|\boldsymbol{A}\|_F$ denote the Frobenius norm of a matrix $\boldsymbol{A}$. The subscript of norm will be dropped when it is clear from the context. $\boldsymbol{a}_{[q]}$ denotes the $q$th element of a vector $\boldsymbol{a}$. $[n]$ denotes the set of integers from 1 to $n$.

### 2.2 Definitions

**Definition 1.** [48] Let $(\Theta, \rho_\Theta)$, $(\mathcal{V}, \rho_\mathcal{V})$ be two metric spaces. A function $h : \Theta \to \mathcal{V}$ is called *Lipschitz continuous* if $\exists L < \infty, \forall \boldsymbol{\theta}_1, \boldsymbol{\theta}_2 \in \Theta$,

$$\rho_\mathcal{V}(h(\boldsymbol{\theta}_1), h(\boldsymbol{\theta}_2)) \leq L\rho_\Theta(\boldsymbol{\theta}_1, \boldsymbol{\theta}_2).$$

The *Lipschitz constant* of $h$ with respect to the input space $\Theta$, denoted by $\mathrm{lip}(h; \mathcal{V} \leftarrow \Theta)$ or $\mathrm{lip}(h \leftarrow \boldsymbol{\theta})$ for short, is the smallest $L$ such that the above inequality holds.

**Definition 2.** A function $r : \Theta \to \mathbb{R}$ is called *Lipschitz smooth*, if $\exists \eta < \infty, \forall \boldsymbol{\theta}_1, \boldsymbol{\theta}_2 \in \Theta$,

$$\|\nabla r(\boldsymbol{\theta}_1) - \nabla r(\boldsymbol{\theta}_2)\| \leq \eta \|\boldsymbol{\theta}_1 - \boldsymbol{\theta}_2\|.$$

The Lipschitz constant of the derivative of $r$ with respect to $\Theta$, denoted by $\mathrm{lip}(\frac{\partial r}{\partial \boldsymbol{\theta}} \leftarrow \boldsymbol{\theta})$, is the smallest $\eta$ such that the above inequality holds.

Some properties of Lipschitz functions will be frequently used in the paper, such as constructing sophisticated Lipschitz functions from the basic ones and bounding the Lipschitz constant via the gradient of differentiable functions; details are listed in Appendix A.

**Definition 3.** [48] The *diameter* of a set $\mathcal{V}$ is defined as

$$\mathrm{diam}(\mathcal{V}) = \max_{\boldsymbol{v}_i, \boldsymbol{v}_j \in \mathcal{V}} \|\boldsymbol{v}_i - \boldsymbol{v}_j\|.$$

**Definition 4.** [43, 22] A hypothesis class $\mathcal{H}$ is *agnostic PAC learnable* if there exist a function $n_{\mathcal{H}}^{AL} : (0,1)^2 \to \mathbb{N}$ and a learning algorithm with the following property: For every $\epsilon, \delta \in (0,1)$ and for every distribution $\mathcal{D}_{\mathcal{Z}}$ over $\mathcal{Z}$, when running the learning algorithm on $n \geq n_{\mathcal{H}}^{AL}(\epsilon, \delta)$ i.i.d. instances generated by $\mathcal{D}_{\mathcal{Z}}$, the algorithm returns a hypothesis $\hat{h}$ such that the following holds:

$$\mathbb{P}_{\boldsymbol{z}^n}\left[R(\hat{h}) - \min_{h \in \mathcal{H}} R(h) \leq \epsilon\right] \geq 1 - \delta.$$

# 3 Learnability via the generalization PAC bound

In this section, we first introduce the generalization PAC bound and establish its link with the agnostic PAC learnability. We then propose a decomposition theorem. Finally, we apply the theorem to prove the learnability of the gradient descent algorithm.

## 3.1 Generalization PAC bound and agnostic PAC learnability

One classical way of determining whether a hypothesis class is agnostic PAC learnable is to verify the uniform convergence condition, which bounds the generalization gap over all hypotheses of the class. However, as some hypotheses are not searched under a fixed setting of the optimization algorithm, [6] proposes to bound the generalization gap for specific algorithms. We adopt this notion and formally define the generalization PAC bound as follows.

**Definition 5.** A hypothesis class $\mathcal{H}$ has the *generalization PAC bound* if there exists a function $n_{\mathcal{H}}^{G} : (0,1)^2 \to \mathbb{N}$ such that for every $\epsilon, \delta \in (0,1)$ and for every probability distribution $\mathcal{D}_{\mathcal{Z}}$ over $\mathcal{Z}$, if $\boldsymbol{z}^n$ is a sample of $n \geq n_{\mathcal{H}}^{G}(\epsilon, \delta)$ i.i.d. examples drawn from $D_{\mathcal{Z}}$, the algorithm returns a hypothesis $\hat{h}$ such that the following inequality is satisfied:

$$\mathbb{P}_{\boldsymbol{z}^n}[R(\hat{h}) - R_n(\boldsymbol{z}^n, \hat{h}) \leq \epsilon] \geq 1 - \delta. \tag{1}$$

First, we note that $\hat{h}$ is regarded as a random variable in this paper. Second, while the generalization PAC bound is a weaker condition than the uniform convergence [6, or Appendix B.2], it is still a sufficient condition for the agnostic PAC learnability, as shown in Theorem 1. Proof of the theorem is given in Appendix B.3.

**Theorem 1.** Suppose $\mathrm{ERM}_{\mathcal{H}}$ exists for a class $\mathcal{H}$, where $\mathrm{ERM}_{\mathcal{H}}$ denotes the empirical risk minimization learner over the class $\mathcal{H}$. If $\mathcal{H}$ has the generalization PAC bound with a function $n_{\mathcal{H}}^{G} : (0,1)^2 \to \mathbb{N}$, then $\mathcal{H}$ is agnostic PAC learnable with the sample complexity function $n_{\mathcal{H}}^{AL}(\epsilon, \delta) \leq \max[n_{\mathcal{H}}^{G}(\epsilon/2, \delta/2), \frac{2C_r^2}{\epsilon^2} \ln \frac{4}{\delta}]$, where the range of the risk function $r(\boldsymbol{z}, h)$ is $[0, C_r]$. Furthermore, in this case, $\mathrm{ERM}_{\mathcal{H}}$ is a successful agnostic PAC learner for $\mathcal{H}$.

## 3.2 Decomposition theorem for the generalization PAC bound

Directly bounding Eq. 1 is difficult due to the random nature of $\boldsymbol{z}^n$ and $\hat{h}$ in $R_n$. To disentangle these two quantities, we propose the following decomposition theorem. Its core idea is to use the uniform convergence bound in a much smaller set.

**Theorem 2** (Decomposition Theorem). Let $\mathcal{W}$ denote the set of all possible values of $\boldsymbol{w}$ and $\hat{\mathcal{W}} \subseteq \mathcal{W}$; let $\delta_1, \delta_2 \geq 0$. If

$$\mathbb{P}_{\boldsymbol{z}^n}[\hat{\boldsymbol{w}} \in \hat{\mathcal{W}}] \geq 1 - \delta_1 \tag{2}$$

and

$$\mathbb{P}_{\boldsymbol{z}^n}\left[\max_{\boldsymbol{w} \in \hat{\mathcal{W}}} \left(R(\boldsymbol{w}) - R_n(\boldsymbol{z}^n, \boldsymbol{w})\right) \leq \epsilon\right] \geq 1 - \delta_2, \tag{3}$$

then the following inequality holds:

$$\mathbb{P}_{\boldsymbol{z}^n}[R(\hat{\boldsymbol{w}}) - R_n(\boldsymbol{z}^n, \hat{\boldsymbol{w}}) \leq \epsilon] \geq 1 - \delta_1 - \delta_2. \tag{4}$$

Recall that $h$ is fully parameterized by $\boldsymbol{w}$ and thus $R(\hat{h})$ and $R_n(\boldsymbol{z}^n, \hat{h})$ are equivalent to $R(\hat{\boldsymbol{w}})$ and $R_n(\boldsymbol{z}^n, \hat{\boldsymbol{w}})$, respectively. Theorem 2 decomposes the generalization PAC bound into two terms which are easier to be bounded, namely a smaller parameter space $\hat{\mathcal{W}}$ that includes estimated parameter vectors with high probability (Eq. 2) and uniform convergence of $\hat{\mathcal{W}}$ (Eq. 3). In the following section, the theorem is applied to analyze the generalization ability of the gradient descent algorithm. We show that Eq. 2 can be guaranteed by applying the concentration inequality to the random variables $\hat{\boldsymbol{w}}$ and Eq. 3 can be guaranteed based on the covering number.

### 3.3 Learnability of the gradient descent algorithm

#### 3.3.1 Settings

The updating equation of the most conventional GD algorithm is as follows:

$$\hat{\boldsymbol{w}}^{(1)} = \boldsymbol{w}^{(0)} - \frac{\alpha^{(1)}}{n} \sum_{i=1}^{n} \frac{\partial r(\boldsymbol{z}_i, \boldsymbol{w})}{\partial \boldsymbol{w}}\Big|_{\hat{\boldsymbol{w}}^{(0)}};$$

$$\vdots$$

$$\hat{\boldsymbol{w}}^{(T)} = \hat{\boldsymbol{w}}^{(T-1)} - \frac{\alpha^{(T)}}{n} \sum_{i=1}^{n} \frac{\partial r(\boldsymbol{z}_i, \boldsymbol{w})}{\partial \boldsymbol{w}}\Big|_{\hat{\boldsymbol{w}}^{(T-1)}}$$

$$= \boldsymbol{w}^{(0)} - \sum_{t=1}^{T} \frac{\alpha^{(t)}}{n} \sum_{i=1}^{n} \frac{\partial r(\boldsymbol{z}_i, \boldsymbol{w})}{\partial \boldsymbol{w}}\Big|_{\hat{\boldsymbol{w}}^{(t-1)}},$$

where $\alpha^{(t)} \geq 0$ denotes the learning rate at iteration $t$; $\hat{\boldsymbol{w}}^{(t)}$ denotes the estimated parameters of the classifier obtained after $t$ iterations; $\boldsymbol{w}^{(0)}$ denotes the initial parameter of the algorithm. Here the number of iterations $T$ and the learning rate $\alpha^{(t)}$ are treated as the setting parameters of the GD algorithm and determined in advance, i.e. $\boldsymbol{s} = \{T, \alpha^{(t)}, t = 1, \ldots, T\}$. The initial weight $\boldsymbol{w}^{(0)}$ is assumed to be fixed.

#### 3.3.2 Concentration of $\hat{\boldsymbol{w}}^{(T)}$

Recall that $\boldsymbol{m}^{(T)}(\boldsymbol{z}^n; \boldsymbol{s}) = \hat{\boldsymbol{w}}^{(T)} \in \mathbb{R}^Q$ and $\boldsymbol{m}_{[q]}^{(T)}(\boldsymbol{z}^n; \boldsymbol{s})$ denotes the $q$th element of $\boldsymbol{m}^{(T)}(\boldsymbol{z}^n; \boldsymbol{s})$. To prove that the first term of Theorem 2 holds, we set $\hat{\mathcal{W}}$ as the Euclidean ball centered at $\mathbb{E}_{\boldsymbol{z}^n} \boldsymbol{m}(\boldsymbol{z}^n; \boldsymbol{s})$ with radius $\epsilon$, denoted by ball$\left(\mathbb{E}_{\boldsymbol{z}^n} \boldsymbol{m}(\boldsymbol{z}^n; \boldsymbol{s}), \epsilon\right)$. The condition that $\hat{\boldsymbol{w}} \in \hat{\mathcal{W}}$ with high probability is equivalent to the condition that $\boldsymbol{m}(\boldsymbol{z}^n; \boldsymbol{s}) \in$ ball$\left(\mathbb{E}_{\boldsymbol{z}^n} \boldsymbol{m}(\boldsymbol{z}^n; \boldsymbol{s}), \epsilon\right)$ with high probability. With a fixed setting $\boldsymbol{s}$ and any fixed initial parameter vector $\boldsymbol{w}^{(0)}$, given the training samples $\boldsymbol{z}^n$, the value of $\boldsymbol{m}_{[q]}^{(T)}(\boldsymbol{z}^n; \boldsymbol{s})$ is determined. In other words, $\boldsymbol{m}_{[q]}^{(T)}(\boldsymbol{z}^n; \boldsymbol{s})$ is a function from $\mathcal{Z}^n$ to $\mathbb{R}$. By applying the McDiarmid's inequality (Lemma B.1), we obtain the following lemma on the concentration property of $\boldsymbol{m}^{(T)}(\boldsymbol{z}^n; \boldsymbol{s})$.

**Lemma 1.** The following bound holds for any fixed $\boldsymbol{s}$ and $\boldsymbol{w}^{(0)}$:

$$\mathbb{P}_{\boldsymbol{z}^n}\left[\boldsymbol{m}^{(T)}(\boldsymbol{z}^n; \boldsymbol{s}) \in \text{ball}\left(\mathbb{E}_{\boldsymbol{z}^n} \boldsymbol{m}^{(T)}(\boldsymbol{z}^n; \boldsymbol{s}), \epsilon\right)\right] \geq 1 - 2Q \exp\left(\frac{-2\epsilon^2 n}{QC^2}\right), \tag{5}$$

where $\boldsymbol{m}^{(T)}(\boldsymbol{z}^n; \boldsymbol{s}) \in \mathbb{R}^Q$; $C = 2\big(\sum_{t=1}^{T} \eta^{T-t}\alpha^{(t)}\big) \mathrm{lip}\,(r \leftarrow \boldsymbol{w})$; $\eta = \max_{i\in[n]} \mathrm{lip}(G_i \leftarrow \boldsymbol{w})$ and $G_i(\boldsymbol{m}^{(t-1)}(\boldsymbol{z}^n)) = \boldsymbol{m}^{(t-1)}(\boldsymbol{z}^n) - \frac{\alpha^{(t)}}{n}\sum_{j\in[n]/i} \frac{\partial r(\boldsymbol{z}_j, \boldsymbol{w})}{\partial \boldsymbol{w}}|_{\boldsymbol{m}^{(t-1)}(\boldsymbol{z}^n)}$; $[n]/i$ denotes the set which contains the integers from $1$ to $n$ without $i$.

The key idea behind the proof is as follows. Randomness of sampling leads to randomness of the learned parameter vector $\hat{\boldsymbol{w}}$. After one iteration of gradient update, the difference between $\hat{\boldsymbol{w}}$ learned on the random samples and that learned on the population is controlled by the Lipschitz constant of $r$ and $G$. Such differences will accumulate over iterations, thereby affecting the concentration property.

### 3.3.3 Uniform convergence inside $\hat{\mathcal{W}}$

The following uniform convergence condition is obtained based on the covering number and Dudley's chaining integral [14]. By using the Lipschitz constant, we can bound the covering number of the hypothesis class by the covering number of the parameter space.

**Lemma 2.** Suppose $\mathrm{lip}(r \leftarrow \boldsymbol{w}) \leq L$ and $\mathrm{diam}(\mathcal{W}) \leq B$, then the following inequality holds:

$$\mathbb{P}_{\boldsymbol{z}^n}\left[ \max_{\boldsymbol{w}\in\mathcal{W}} \big(R(\boldsymbol{w}) - R_n(\boldsymbol{z}^n, \boldsymbol{w})\big) \leq CLB\sqrt{\frac{Q}{n}} + \sqrt{\frac{\ln(1/\delta)}{2n}}\right] \geq 1 - \delta, \qquad (6)$$

where $C$ is a universal constant.

### 3.3.4 Application of the decomposition theorem

**Theorem 3.** Suppose $\mathrm{lip}(h \leftarrow \boldsymbol{w}) \leq L_1$ and $\mathrm{lip}(l \leftarrow h) \leq L_l$. Then with probability at least $1 - \delta_1 - \delta_2$, the following bound holds:

$$R(\boldsymbol{m}(\boldsymbol{z}^n; \boldsymbol{s})) - R_n(\boldsymbol{z}^n, \boldsymbol{m}(\boldsymbol{z}^n; \boldsymbol{s})) \leq \frac{C_1 C_2 L_1^2 L_l^2 Q \sqrt{\ln(2Q/\delta_1)}}{n} + \sqrt{\frac{\ln(1/\delta_2)}{2n}}, \qquad (7)$$

where $\boldsymbol{w} \in \mathbb{R}^Q$; $C_1$ is a universal constant; $C_2 = \sum_{t=1}^{T} \eta^{T-t}\alpha^{(t)}$, in which $T$ denotes the number of iterations, $\alpha^{(t)}$ denotes the learning rate at iteration $t$, $\eta = \max_{i\in[n]} \mathrm{lip}(G_i \leftarrow \boldsymbol{w})$, and $G_i(\boldsymbol{m}^{(t-1)}(\boldsymbol{z}^n)) = \boldsymbol{m}^{(t-1)}(\boldsymbol{z}^n) - \frac{\alpha^{(t)}}{n}\sum_{j\in[n]/i} \frac{\partial r(\boldsymbol{z}_j, \boldsymbol{w})}{\partial \boldsymbol{w}}|_{\boldsymbol{m}^{(t-1)}(\boldsymbol{z}^n)}$; $[n]/i$ denotes the set which contains the integers from $1$ to $n$ without $i$.

Theorem 3 suggests that the following factors will affect the generalizability of the learned model.
1) $T$: A smaller number of training iterations leads to better concentration property and thus better generalization performance. Thus, when optimizing via GD, we shall select the model from the earliest iteration $t$ that yields the minimum training error; the test stage is implemented using the parameters learned at $t$;
2) $Q$: A smaller value of $Q$, i.e. fewer parameters, gives a tighter generalization bound;
3) $L_1, L_l$: Using a classifier and loss function with smaller Lipschitz constants will improve the generalizability;
4) $\eta$: Based on the definition of $G$ and the addition property of Lipschitz functions (Appendix A), if $\mathrm{lip}(\frac{\partial r(\boldsymbol{z}_j, \boldsymbol{w})}{\partial \boldsymbol{w}} \leftarrow \boldsymbol{w})$ is bounded by $L_s$, then $\eta$ is bounded by $1 + \alpha L_s$, where $\alpha = \max_{t\in[T]} \alpha^{(t)}$. Based on the composition property of Lipschitz functions, we have

$$\mathrm{lip}(\frac{\partial r}{\partial \boldsymbol{w}} \leftarrow \boldsymbol{w}) = \mathrm{lip}(\frac{\partial l}{\partial \boldsymbol{w}} \leftarrow \boldsymbol{w}) = \mathrm{lip}(\frac{\partial l}{\partial h}\frac{\partial h}{\partial \boldsymbol{w}} \leftarrow \boldsymbol{w}) \leq \mathrm{lip}(\frac{\partial l}{\partial h} \leftarrow h)\,\mathrm{lip}(\frac{\partial h}{\partial \boldsymbol{w}} \leftarrow \boldsymbol{w}).$$

Thus $\eta$ is bounded if both $\mathrm{lip}(\frac{\partial h}{\partial \boldsymbol{w}} \leftarrow \boldsymbol{w})$ and $\mathrm{lip}(\frac{\partial l}{\partial h} \leftarrow h)$ are bounded. In other words, the classifier and loss function should be Lipschitz smooth.

We make a final remark that, since the decomposition theorem (Theorem 2) and the concentration lemma (Lemma 1) are established regardless of the number of classes, the proposed generalization PAC bound can be readily generalized to multi-class classification. The only modification required is to establish that uniform convergence holds for the hypothesis class with a multi-class loss function $l$, which can be proved based on, e.g., VC-dimension [1] and Rademacher complexity [34, 30].

# 4 Smooth metric and representative instance learning (SMILE)

Theorem 3 shows that Lipschitz smoothness is important for ensuring generalization. To enjoy and illustrate the practical exploitation of this appealing theoretical result, we establish a simple yet theoretically well-founded and new metric learning method called SMILE with a smooth classifier and a smooth loss function. SMILE learns a Mahalanobis distance to enhance the classification performance of NN classifier. Meanwhile, to reduce the storage and computational cost of NN, SMILE learns few representative instances in the training stage and calculate the distances between the test instance and representative instances only in the test stage. In this section, we present the classifier, the loss function, the optimization problem, and some experimental results of SMILE.

## 4.1 The classifier of SMILE

For any two instances $\boldsymbol{x}_i$ and $\boldsymbol{x}_j$, the generalized Mahalanobis distance is defined as $d_{\boldsymbol{M}}(\boldsymbol{x}_i, \boldsymbol{x}_j) = \sqrt{(\boldsymbol{x}_i - \boldsymbol{x}_j)^T \boldsymbol{M}(\boldsymbol{x}_i - \boldsymbol{x}_j)}$, where $\boldsymbol{M}$ is a positive semidefinite (PSD) matrix. Owing to the PSD property, $\boldsymbol{M} = \boldsymbol{L}^T \boldsymbol{L}$ and thus $d_{\boldsymbol{M}}(\boldsymbol{x}_i, \boldsymbol{x}_j) = d(\boldsymbol{L}\boldsymbol{x}_i, \boldsymbol{L}\boldsymbol{x}_j) = \|\boldsymbol{L}\boldsymbol{x}_i - \boldsymbol{L}\boldsymbol{x}_j\|_2$.

The classifier of SMILE is simply defined as follows:

$$h(\boldsymbol{x}; \boldsymbol{r}^m, \boldsymbol{L}) = \sum_j \exp(-d^2(\boldsymbol{L}\boldsymbol{x}, \boldsymbol{r}_j^+)) - \sum_k \exp(-d^2(\boldsymbol{L}\boldsymbol{x}, \boldsymbol{r}_k^-)), \tag{8}$$

where $\boldsymbol{r}^m$ and $\boldsymbol{L}$ are the parameters of the classifier; $\boldsymbol{r}_j^+$ and $\boldsymbol{r}_k^-$ denote the $j$th representative instance of the positive class and the $k$th representative instance of the negative class, respectively; $m$ denotes the total number of learned representative instances. The test instance $\boldsymbol{x}$ is classified to the positive class when $h(\boldsymbol{x}) \geq 0$ and to the negative class when $h(\boldsymbol{x}) < 0$.

As shown in Appendix C, a sufficient condition for $h$ to be Lipschitz smooth is that $\mathrm{diam}(\boldsymbol{L})$, $\mathrm{diam}(\boldsymbol{x})$ and $\mathrm{diam}(\boldsymbol{r})$ are bounded. With a slight abuse of notation, $\mathrm{diam}(\boldsymbol{L})$ denotes the diameter of the set which contains all possible values of $\boldsymbol{L}$; $\mathrm{diam}(\boldsymbol{x})$ and $\mathrm{diam}(\boldsymbol{r})$ are defined similarly. To bound these quantities, we will constrain the Frobenius norm of $\boldsymbol{L}$ and the $L_2$-norm of $\boldsymbol{x}$ and $\boldsymbol{r}$.

## 4.2 The loss function of SMILE

Similarly to the Huber loss for regression [25], we propose the following loss function defined by combining a quadratic and a linear function:

$$l(a) = \begin{cases} 1 - a & \text{if } a \leq 0; \\ \frac{1}{4}(a-2)^2 & \text{if } 0 < a \leq 2; \\ 0 & \text{if } a > 2. \end{cases} \tag{9}$$

The derivative of $l(a)$ is as follows:

$$l'(a) = \begin{cases} -1 & \text{if } a \leq 0; \\ \frac{a-2}{2} & \text{if } 0 < a \leq 2; \\ 0 & \text{if } a > 2. \end{cases}$$

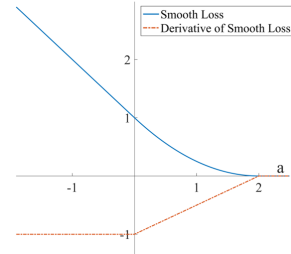

Figure 1: Illustration of the proposed Lipschitz smooth loss function and its derivative.

The loss function and its derivative are illustrated in Figure 1. The Lipschitz constant of $l'(a)$ is $\frac{1}{2}$, meaning that the proposed loss is a Lipschitz smooth function.

## 4.3 The optimization problem of SMILE

Using the classifier defined in Eq. 8, the loss function defined in Eq. 9, and the convex regularization terms $\sum_j \|\boldsymbol{r}_j^+\|_2^2 + \sum_k \|\boldsymbol{r}_k^-\|_2^2 + \|\boldsymbol{L}\|_F^2$, the following optimization problem is proposed for SMILE:

$$\min_{\Theta} \frac{1}{n} \sum_{i=1}^{n} l(y_i h(\boldsymbol{x}_i; \boldsymbol{r}^m, \boldsymbol{L})) + \lambda \Big( \sum_{j=1}^{m_+} \|\boldsymbol{r}_j^+\|_2^2 + \sum_{k=1}^{m_-} \|\boldsymbol{r}_k^-\|_2^2 + \|\boldsymbol{L}\|_F^2 \Big), \tag{10}$$

where $\Theta = \{\boldsymbol{r}^m, \boldsymbol{L}\}$ denotes the set of parameters to be optimized; $\boldsymbol{r}^m = \{\boldsymbol{r}_j^+, \boldsymbol{r}_k^-; j = 1, \ldots, m_+, k = 1, \ldots, m_-\}$ denotes the set of representative instances with $m_+$ instances for

Table 1: Comparison of classification performances. Mean accuracy and standard deviations are reported with the best ones in bold; '# of best' denotes the number of datasets on which the metric learning algorithm obtains the highest accuracy.

| Dataset | NCA | LMNN | ITML | R2LML | RVML | GMML | DMLMJ | DMPL | SNC | SMILE |
|---|---|---|---|---|---|---|---|---|---|---|
| Australian | 80.0±1.6 | 78.8±2.6 | 77.2±1.9 | 84.7±1.3 | 83.0±1.6 | 84.4±1.0 | 83.9±1.3 | 84.8±1.5 | 81.8±8.8 | **86.0±0.7** |
| Cancer | 95.4±1.3 | 96.0±0.7 | 96.1±1.1 | 96.7±0.8 | 95.2±1.0 | 96.5±0.8 | 96.5±0.5 | **96.8±0.6** | 95.1±1.7 | **96.8±0.6** |
| Climate | 91.5±2.1 | 91.8±1.3 | 86.7±1.0 | 91.7±1.7 | 92.2±1.1 | 91.3±2.5 | 92.9±1.9 | **93.6±2.0** | 92.0±1.7 | 93.5±1.7 |
| Credit | 80.6±2.0 | 82.2±1.4 | 77.6±2.0 | **86.1±1.5** | 83.5±1.8 | 85.9±1.7 | 84.6±1.4 | 85.5±1.7 | 83.4±3.7 | 85.6±1.9 |
| German | 70.0±2.9 | 67.9±1.5 | 67.0±2.1 | 72.9±1.8 | 71.7±1.8 | 71.6±1.1 | 69.3±2.7 | 71.7±2.3 | 70.1±3.3 | **75.5±1.1** |
| Haberman | 67.4±3.3 | 67.9±3.3 | 68.0±4.1 | 71.1±3.4 | 66.7±2.3 | 71.2±3.4 | 68.5±3.2 | 69.9±3.3 | 72.0±5.2 | **72.4±3.3** |
| Heart | 75.6±2.0 | 76.2±3.8 | 76.9±3.3 | 82.0±3.8 | 77.7±4.1 | 81.2±2.7 | 80.6±2.8 | 77.9±3.4 | 77.0±5.3 | **84.0±2.2** |
| ILPD | 66.8±1.2 | 67.0±2.1 | 68.7±2.8 | 65.9±2.2 | 68.0±2.9 | 67.1±2.2 | 68.0±1.6 | 68.1±2.3 | 68.9±2.7 | **71.3±1.7** |
| Liver | 59.8±3.4 | 61.0±4.8 | 57.2±4.0 | **66.8±3.7** | 64.6±3.9 | 63.8±5.4 | 60.9±3.8 | 62.2±7.6 | 63.3±5.2 | 62.8±5.8 |
| Pima | 65.9±3.0 | 68.5±1.6 | 68.0±2.0 | 72.3±1.5 | 69.5±1.7 | 73.0±1.8 | 71.1±2.3 | 71.0±2.8 | **74.0±2.6** | 73.2±2.0 |
| Ringnorm | 69.3±0.7 | 65.2±0.7 | 65.8±0.9 | NA | 72.3±0.6 | 72.5±0.5 | 73.9±0.7 | NA | 71.3±0.6 | **77.1±0.5** |
| Twonorm | 96.7±0.4 | 95.6±0.5 | 96.4±0.3 | NA | 97.3±0.3 | 97.5±0.3 | 97.7±0.2 | NA | 97.3±0.2 | **97.9±0.3** |
| Average | 76.6 | 76.5 | 75.5 | NA | 78.1 | 78.5 | 79.7 | NA | 79.0 | **81.3** |
| # of best | 0 | 0 | 0 | 2 | 0 | 0 | 0 | 2 | 1 | **8** |

the positive class and $m_-$ instances for the negative class; and $\lambda$ is a trade-off parameter balancing the loss term and the regularization term.

The objective function is not convex due to the non-convexity of $h(\boldsymbol{x}; \boldsymbol{r}^m, \boldsymbol{L})$. We apply the gradient descent algorithm to learn the parameters; detailed formulae are given in Appendix C.5. The generalization guarantee for SMILE is provided in Appendix C.4.

## 4.4 Illustrative results of SMILE

**Experimental settings**  We illustrate the effectiveness of SMILE by comparing it with nine widely adopted metric learning methods: NCA [17], LMNN [49], ITML [10], R2LML [23], RVML [39], GMML [56], DMLMJ [37], DMPL [28], and SNC [29]; the last two methods learn both the metric and representative instances. NCA is implemented by using the drToolbox [45]; LMNN and ITML are implemented by using the metric-learn toolbox [11]; and R2LML, RVML, GMML, DMLMJ, and SNC are implemented by using the authors' code.

The experiment focuses on binary classification of 12 publicly available datasets from the websites of UCI [13] and Delve [41]. Sample size and feature dimension are listed in Table 1 of Appendix D. All datasets are pre-processed by firstly subtracting the mean and dividing by the standard deviation, and then normalizing the $L_2$-norm of each instance to 1.

For each dataset, we randomly select $60\%$ of instances to form a training set and the rest are used for testing. This process is repeated 10 times and we report the mean accuracy and the standard deviation. 10-fold cross-validation is used to select the trade-off parameters in the compared algorithms, namely the regularization parameter $\mu$ in LMNN (from $\{0.1, 0.3, \ldots, 0.9\}$), $\gamma$ in ITML (from $\{0.25, 0.5, 1, 2, 4\}$), $\lambda$ in RVML (from $\{10^{-5}, 10^{-4}, \ldots, 10\}$), $t$ in GMML (from $\{0.1, 0.3, \ldots, 0.9\}$), $\mu$ in DMPL (from $\{0.1, 0.3, \ldots, 0.9\}$), and ratio in SNC (from $\{0.01, 0.02, 0.04, 0.08, 0.16\}$). All other parameters are set as default. For the proposed SMILE, the parameters are set as follows: $\boldsymbol{L}$ is initialized as the identity matrix; $\boldsymbol{r}^m$ are initialized as the $k$-means clustering centers of the positive and negative classes (by using MATLAB *kmeans* function with random initial values); the number of representative instances for each class is set as 2; the trade-off parameter $\lambda$ is set as 1; and the learning rate $\alpha$ is set as 0.001. The maximum number of iterations is set as 5000 and the final result is based on the parameters at time $t$, which is the earliest time when the smallest training error is obtained, to conform to early stopping as suggested by Theorem 3. Code for the proposed method is available at http://github.com/xyang6/SMILE.

**Evaluation on classification performance**  As shown in Table 1, with only two representative instances learned for each class, the proposed SMILE achieves the best accuracy on 8 out of the 12 datasets; none of the other methods performs the best on more than 2 datasets. The average accuracy of SMILE is also the highest. These results suggest that SMILE, although simple, enjoys competitive performance against existing metric learning algorithms, thanks to its theoretical foundation.

**Visualization of the concentration behavior**   Our theoretical finding suggests that randomness of parameters is caused by random sampling and will accumulate over iterations. We now verify this finding with an empirical study on the German dataset. More specifically, we learn parameters $\boldsymbol{L}, \boldsymbol{r}^m$ from a subset of the data, which serves as $\boldsymbol{m}^{(T)}(\boldsymbol{z}^n)$ in Lemma 1, learn parameters from the entire dataset, which serves as $\mathbb{E}_{\boldsymbol{z}^n}\boldsymbol{m}^{(T)}(\boldsymbol{z}^n)$, and quantify their differences via the $L_2$-norm. The total sample size is 1000 and the subset size is selected as $\{100, 200, \ldots, 500\}$. After randomly sampling the subset for 100 times, we calculate the 95th percentile of the norm differences and denote this value as $\epsilon_{95\%}$. $\epsilon_{95\%}$ can be inter-

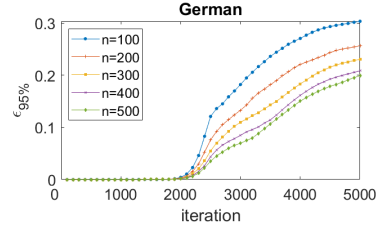

Figure 2: Effect of training iterations and sample size on parameter concentration.

preted as the minimum radius $\epsilon$ of ball $\left(\mathbb{E}_{\boldsymbol{z}^n}\boldsymbol{m}^{(T)}(\boldsymbol{z}^n), \epsilon\right)$ such that the bound (Eq. 5) holds with 95% probability. From Figure 2, we first see that learning from fewer training instances leads to a larger value of $\epsilon_{95\%}$, which signifies that sampling randomness contributes to the variance of learned parameters. Second, we see that learning with more iterations increases $\epsilon_{95\%}$, which is also consistent with the theoretical result. Moreover, the rate of increase is exponential in the early stage of training and decreases gradually towards zero, which implies that parameters are optimized to local minima and will no longer be updated.

**Analysis of the effect of parameter** $\lambda$   Figure 3 illustrates the impact of the trade-off parameter $\lambda$ in Eq. 10 on the generalization gap, training accuracy and test accuracy; the Heart dataset is used as an example. The left-hand figure shows that the generalization gap decreases with $\lambda$. This is consistent with our theoretical result that constraining the norms of $\boldsymbol{L}, \boldsymbol{x}$ and $\boldsymbol{r}$ gives smaller Lipschitz constants, thereby tightening the bound. The right-hand figure shows that, as the training accuracy generally decreases with $\lambda$ as well, the test accuracy is highest when $\lambda = 0.5$.

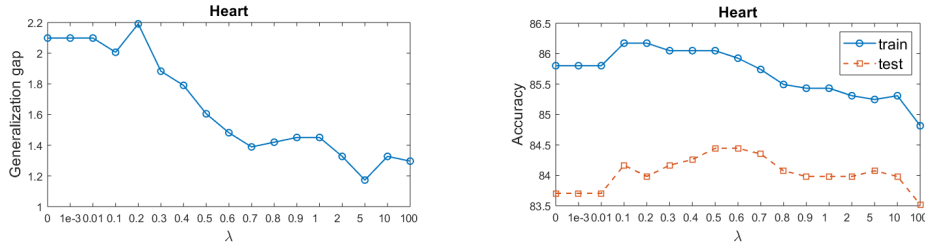

Figure 3: Effect of $\lambda$ on the generalization gap (*left*) and the training or test accuracy (*right*).

# 5   Conclusion

This paper presents a new route to the generalization guarantee on classifiers optimized via GD, considering the influence of sampling randomness to the concentration property of parameters and embracing algorithmic parameters. We propose a new decomposition theorem to obtain the generalization PAC bound, which consequently guarantees the agnostic PAC learnability. We demonstrate the importance of Lipschitz smooth classifiers and loss functions for generalization and theoretically justify the benefit of early stopping. Our results are derived based only on the Lipschitz property over the parameter space and hence are applicable to non-convex optimization problems. In addition, we propose a new metric learning method as an illustrative example to demonstrate the practicability of the derived appealing theoretical results.

The generalization PAC bound is only derived for GD and cannot be used to guarantee the learnability of classifiers optimized through stochastic GD and its variants, due to additional randomness of training instances introduced in each mini-batch. It would be valuable to extend our work to these algorithms given their importance in large-scale optimization problems. Moreover, as Figure 2 suggests that the radius used to cover parameters with high confidence expands at a much slower rate after training with a large number of iterations, we intend to investigate the link between this local convergence behavior and the concentration property, and take it into account to derive tighter bounds.

## Broader Impact

This paper is a theoretical analysis relating to gradient descent and metric learning algorithms, making no direct impact on ethical and societal issues. The findings can be used to design more effective training strategies or algorithms, and consequently benefit the downstream applications.

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
