[Supplementary Material · Generalization Bound of Gradient Descent for Non-Convex Metric Learning_supplement.pdf]

# Supplementary Material of Generalization Bound of Gradient Descent for Non-Convex Metric Learning

**Mingzhi Dong**[1,2,3*]   **Xiaochen Yang**[3*]   **Rui Zhu**[4]   **Yujiang Wang**[5]   **Jing-Hao Xue**[3]

[1]School of Computer Science, Fudan University, Shanghai, China
[2]Shanghai Key Laboratory of Data Science, Fudan University, Shanghai, China
[3]Department of Statistical Science, University College London, London, UK
[4]The Business School (formerly Cass), City, University of London, London, UK
[5]Department of Computing, Imperial College London, London, UK

`mingzhidong@gmail.com`   `{xiaochen.yang.16, jinghao.xue}@ucl.ac.uk`
`rui.zhu@city.ac.uk`   `yujiang.wang14@imperial.ac.uk`

## Contents

# A Properties of Lipschitz functions

The Lipschitz constant of differentiable functions can be obtained from their gradients; this follows from the mean value theorem as shown below.

**Theorem A.1.** [4] Let $\mathcal{U} \in \mathbb{R}^n$ be open, $h : \mathcal{U} \to \mathbb{R}$ be differentiable and the line segment $[\boldsymbol{u}_1, \boldsymbol{u}_2] \in \mathcal{U}$, where $[\boldsymbol{u}_1, \boldsymbol{u}_2] = \{\boldsymbol{v} \mid \boldsymbol{v} = \boldsymbol{u}_1 + t(\boldsymbol{u}_2 - \boldsymbol{u}_1), t \in [0,1]\}$ joins $\boldsymbol{u}_1$ to $\boldsymbol{u}_2$. Based on the *Mean Value Theorem*, there exists a $\boldsymbol{u} \in [\boldsymbol{u}_1, \boldsymbol{u}_2]$

$$f(\boldsymbol{u}_2) - f(\boldsymbol{u}_1) = f'(\boldsymbol{u})^T (\boldsymbol{u}_2 - \boldsymbol{u}_1).$$

**Corollary A.1.** Let $\mathcal{U} \in \mathbb{R}^n$ be open and convex, $h : \mathcal{U} \to \mathbb{R}$ be differentiable inside $\mathcal{U}$, then the following inequality holds:

$$\text{lip}(h \leftarrow \boldsymbol{u}) = \max_{\boldsymbol{u}_1, \boldsymbol{u}_2 \in \mathcal{U}, \boldsymbol{u}_1 \neq \boldsymbol{u}_2} \frac{|h(\boldsymbol{u}_2) - h(\boldsymbol{u}_1)|}{\|\boldsymbol{u}_2 - \boldsymbol{u}_1\|} \leq \max_{u \in \mathcal{U}} \|h'(\boldsymbol{u})\|.$$

*Proof.* Since $\mathcal{U}$ is convex, $\forall \boldsymbol{u}_1, \boldsymbol{u}_2 \in \mathcal{U}, \boldsymbol{u}_1 \neq \boldsymbol{u}_2$, the line segment $[\boldsymbol{u}_1, \boldsymbol{u}_2] = \{\boldsymbol{v} \mid \boldsymbol{v} = \boldsymbol{u}_1 + t(\boldsymbol{u}_2 - \boldsymbol{u}_1), t \in [0,1]\} \in \mathcal{U}$.

$$|h(\boldsymbol{u}_2) - h(\boldsymbol{u}_1)| =_{(a)} |h'(\boldsymbol{u})^T (\boldsymbol{u}_2 - \boldsymbol{u}_1)| \leq_{(b)} \|h'(\boldsymbol{u})\|\|\boldsymbol{u}_2 - \boldsymbol{u}_1\| \leq_{(c)} \max_{u \in \mathcal{U}} \|h'(\boldsymbol{u})\|\|\boldsymbol{u}_2 - \boldsymbol{u}_1\|,$$

where equality (a) is due to Theorem A.1; inequality (b) is due to the Cauchy-Schwarz inequality; inequality (c) is due to $\|h'(\boldsymbol{u})\| \leq \max_{u \in \mathcal{U}} \|h'(\boldsymbol{u})\|$.  □

Sophisticated Lipschitz functions can be constructed from the basic ones using the following lemma.

**Lemma A.1.** [5, 9] Let $\text{lip}(h_1 \leftarrow \boldsymbol{u}) \leq L_1$, $\text{lip}(h_2 \leftarrow \boldsymbol{u}) \leq L_2$ and $\text{lip}(h_2 \circ h_1 \leftarrow h_1) \leq L_3$, where $\circ$ denotes the composition of functions. Then
(a) $\text{lip}(ah_1 \leftarrow \boldsymbol{u}) \leq |a|L_1$, where $a$ is a constant;
(b) $\text{lip}(h_1 + h_2 \leftarrow \boldsymbol{u}) \leq L_1 + L_2$, $\text{lip}(h_1 - h_2 \leftarrow \boldsymbol{u}) \leq L_1 + L_2$;
(c) $\text{lip}(\min(h_1, h_2) \leftarrow \boldsymbol{u}) \leq \max\{L_1, L_2\}$, $\text{lip}(\max(h_1, h_2) \leftarrow \boldsymbol{u}) \leq \max\{L_1, L_2\}$, where $\min(h_1, h_2)$ or $\max(h_1, h_2)$ denote the pointwise minimum or maximum of functions $h_1$ and $h_2$;
(d) $\text{lip}(h_2 \circ h_1 \leftarrow \boldsymbol{u}) \leq L_1 L_3$.

This lemma illustrates that after the operations of multiplication by constant, addition, subtraction, minimization, maximization and function composition, the functions are still Lipschitz continuous.

# B Proofs of theorems and lemmas

## B.1 Preliminary

The following McDiarmid's inequality will be frequently used in subsequent proofs.

**Lemma B.1.** [6] Let $\boldsymbol{z}^n = \{\boldsymbol{z}_1, \ldots, \boldsymbol{z}_{i-1}, \boldsymbol{z}_i, \boldsymbol{z}_{i+1}, \ldots, \boldsymbol{z}_n\}$ be $n$ independent samples. Let $\boldsymbol{z}^{n,i} = \{\boldsymbol{z}_1, \ldots, \boldsymbol{z}_{i-1}, \boldsymbol{z}_i', \boldsymbol{z}_{i+1}, \ldots, \boldsymbol{z}_n\}$, where the replacement example $\boldsymbol{z}_i'$ is assumed to be drawn from the same distribution of $\boldsymbol{z}_i$ and is independent from $\boldsymbol{z}^n$. Furthermore, let $m : \mathcal{Z}^n \to \mathbb{R}$ be a function of $\boldsymbol{z}_1, \ldots, \boldsymbol{z}_n$ that satisfies $\forall i, \forall \boldsymbol{z}^n, \forall \boldsymbol{z}^{n,i}$

$$|m(\boldsymbol{z}^n) - m(\boldsymbol{z}^{n,i})| \leq c_i, \tag{1}$$

for some constant $c_i$. Then for all $\epsilon > 0$,

$$\mathbb{P}_{\boldsymbol{z}^n}[m(\boldsymbol{z}^n) - \mathbb{E}_{\boldsymbol{z}^n}[m(\boldsymbol{z}^n)] \geq \epsilon] \leq \exp\left(\frac{-2\epsilon^2}{\sum_{i=1}^n c_i^2}\right),$$

$$\mathbb{P}_{\boldsymbol{z}^n}[\mathbb{E}_{\boldsymbol{z}^n}[m(\boldsymbol{z}^n)] - m(\boldsymbol{z}^n) \geq \epsilon] \leq \exp\left(\frac{-2\epsilon^2}{\sum_{i=1}^n c_i^2}\right);$$

that is,

$$\mathbb{P}_{\boldsymbol{z}^n}[|m(\boldsymbol{z}^n) - \mathbb{E}_{\boldsymbol{z}^n}[m(\boldsymbol{z}^n)]| \geq \epsilon] \leq 2\exp\left(\frac{-2\epsilon^2}{\sum_{i=1}^n c_i^2}\right). \tag{2}$$

## B.2 Proof: generalization PAC bound is a weaker condition than uniform convergence

For completeness, we show that the generalization PAC bound is a weaker condition than the uniform convergence bound [1] in Lemma B.2.

**Lemma B.2.** The relationship between the generalization PAC bound and the uniform convergence bound is as follows:

$$\mathbb{P}_{\boldsymbol{z}^n}[R(\hat{h}) - R_n(\boldsymbol{z}^n, \hat{h}) \le \epsilon] \ge \mathbb{P}_{\boldsymbol{z}^n}\Big[\max_{h\in\mathcal{H}}\big(R(h) - R_n(\boldsymbol{z}^n, h)\big) \le \epsilon\Big]. \tag{3}$$

*Proof.* Let $E_1$ be the set of events of $R(\hat{h}) - R_n(\boldsymbol{z}^n, \hat{h}) \le \epsilon$ and $E_2$ be the set of events of $\max_{h\in\mathcal{H}}\big(R(h) - R_n(\boldsymbol{z}^n, h)\big) \le \epsilon$. The probabilities of these two events are given as follows:

$$\mathbb{P}_{\boldsymbol{z}^n}(E_1) = \int \big(p(\boldsymbol{z}^n)\mathbb{1}[E_1]\big)\, d\boldsymbol{z}^n$$

$$\mathbb{P}_{\boldsymbol{z}^n}(E_2) = \int \big(p(\boldsymbol{z}^n)\mathbb{1}[E_2]\big)\, d\boldsymbol{z}^n.$$

1. At the points $\boldsymbol{z}^n$ where $\mathbb{1}[E_2] = 1$, we have $\mathbb{1}[E_1] = 1$ and thus

$$p(\boldsymbol{z}^n)\mathbb{1}[E_1] = p(\boldsymbol{z}^n)\mathbb{1}[E_2].$$

2. At the points $\boldsymbol{z}^n$ where $\mathbb{1}[E_2] = 0$, we have

$$p(\boldsymbol{z}^n)\mathbb{1}[E_1] \ge 0 = p(\boldsymbol{z}^n)\mathbb{1}[E_2].$$

Therefore, integrating over all possible points $\boldsymbol{z}^n$, we have $\int (p(\boldsymbol{z}^n)\mathbb{1}[E_1])d\boldsymbol{z}^n \ge \int (p(\boldsymbol{z}^n)\mathbb{1}[E_2]) d\boldsymbol{z}^n$. That is, $\mathbb{P}_{\boldsymbol{z}^n}(E_1) \ge \mathbb{P}_{\boldsymbol{z}^n}(E_2)$. □

## B.3 Proof of Theorem 1

After proving Proposition B.1, Theorem 1 is proved.

**Proposition B.1.** Suppose the range of the risk function $r(\boldsymbol{z}, h)$ is $[0, C_r]$, then

$$\mathbb{P}_{\boldsymbol{z}^n}\Big[\min_{h\in\mathcal{H}} R_n(\boldsymbol{z}^n, h) - E_{\boldsymbol{z}^n}\big[\min_{h\in\mathcal{H}} R_n(\boldsymbol{z}^n, h)\big] \ge \epsilon\Big) \le \exp\left(\frac{-2n\epsilon^2}{C_r^2}\right).$$

*Proof.* Given $\boldsymbol{z}^n$ and a fixed hypothesis class of $\mathcal{H}$, the value of $a(\boldsymbol{z}^n) = \min_{h\in\mathcal{H}} R_n(\boldsymbol{z}^n, h)$ is fixed and the mapping $a : \mathcal{Z}^n \to \mathbb{R}$ is a function. Therefore, the McDiarmid's inequality (Lemma B.1) can be applied as long as the bounded difference condition (Eq. 1) holds. We show that $|\min_{h\in\mathcal{H}} R_n(\boldsymbol{z}^n, h) - \min_{h\in\mathcal{H}} R_n(\boldsymbol{z}^{n,i}, h)|$ is bounded as follows:

$$\min_{h\in\mathcal{H}} R_n(\boldsymbol{z}^{n,i}, h)$$
$$= \min_{h\in\mathcal{H}}\left(R_n(\boldsymbol{z}^n, h) - \frac{r(z_i, h)}{n} + \frac{r(z_i', h)}{n}\right)$$
$$\le \min_{h\in\mathcal{H}}\left(R_n(\boldsymbol{z}^n, h) - 0 + \frac{C_r}{n}\right)$$
$$= \min_{h\in\mathcal{H}} R_n(\boldsymbol{z}^n, h) + \frac{C_r}{n}.$$

Similarly,

$$\min_{h\in\mathcal{H}} R_n(\boldsymbol{z}^n, h) \le \min_{h\in\mathcal{H}} R_n(\boldsymbol{z}^{n,i}, h) + \frac{C_r}{n}.$$

Therefore

$$\left|\min_{h\in\mathcal{H}} R_n(\boldsymbol{z}^n, h) - \min_{h\in\mathcal{H}} R_n(\boldsymbol{z}^{n,i}, h)\right| \le \frac{C_r}{n}.$$

The result is obtained by substituting $c_i = \frac{C_r}{n}$ into Lemma B.1. □

Theorem 1 is proved as follows.

*Proof.* Let $\hat{h} \in \operatorname{argmin}_{h \in \mathcal{H}} R_n(\boldsymbol{z}^n, h)$, we have

$$R_n(\boldsymbol{z}^n, \hat{h}) = \min_{h \in \mathcal{H}} R_n(\boldsymbol{z}^n, h).$$

Suppose

$$\mathbb{P}_{\boldsymbol{z}^n}[R(\hat{h}) - R_n(\boldsymbol{z}^n, \hat{h}) \leq \epsilon/2] \geq 1 - \delta/2,$$
$$\mathbb{P}_{\boldsymbol{z}^n}\left[R_n(\boldsymbol{z}^n, \hat{h}) - E_{\boldsymbol{z}^n}[R_n(\boldsymbol{z}^n, \hat{h})] \leq \epsilon/2\right] \geq 1 - \delta/2.$$

Let $E_1 = \{\boldsymbol{z}^n | R(\hat{h}) - R_n(\boldsymbol{z}^n, \hat{h}) \leq \epsilon/2\}$ and $E_2 = \{\boldsymbol{z}^n | R_n(\boldsymbol{z}^n, \hat{h}) - E_{\boldsymbol{z}^n}[R_n(\boldsymbol{z}^n, \hat{h})] \leq \epsilon/2\}$. $\forall \boldsymbol{z}^n \in E_1 \cap E_2$, we have

$$R(\hat{h})$$

$$(a) \quad \leq R_n(\boldsymbol{z}^n, \hat{h}) + \frac{\epsilon}{2}$$

$$(b) \quad \leq \mathbb{E}_{\boldsymbol{z}^n}[R_n(\boldsymbol{z}^n, \hat{h})] + \epsilon$$

$$(c) \quad = \mathbb{E}_{\boldsymbol{z}^n} \min_{h \in \mathcal{H}} \frac{\sum_{i=1}^n r(\boldsymbol{z}_i, h)}{n} + \epsilon$$

$$(d) \quad \leq \min_{h \in \mathcal{H}} \mathbb{E}_{\boldsymbol{z}^n} \frac{\sum_{i=1}^n r(\boldsymbol{z}_i, h)}{n} + \epsilon$$

$$(e) \quad = \min_{h \in \mathcal{H}} \mathbb{E}_{\boldsymbol{z}} r(\boldsymbol{z}, h) + \epsilon$$

$$(f) \quad = \min_{h \in \mathcal{H}} R(h) + \epsilon,$$

where inequality (a) is due to $R(\hat{h}) - R_n(\boldsymbol{z}^n, \hat{h}) \leq \epsilon/2$; inequality (b) is due to $R_n(\boldsymbol{z}^n, \hat{h}) - E_{\boldsymbol{z}^n}[R_n(\boldsymbol{z}^n, \hat{h})] \leq \epsilon/2$; equality (c) is due to the definitions of $R_n(\boldsymbol{z}^n, h)$ and $\hat{h}$; inequality (d) is due to change the order of $E_{\boldsymbol{z}^n}$ and $\min_{h \in \mathcal{H}}$; equality (e) is due to the identical assumption of $\boldsymbol{z}^n$; equality (f) is due to the definition of $R(h)$.

Therefore,

$$\mathbb{P}_{\boldsymbol{z}^n}\left[R(\hat{h}) \leq \min_{h \in \mathcal{H}} R(h) + \epsilon\right]$$

$$(a) \quad \geq \mathbb{P}_{\boldsymbol{z}^n}[\boldsymbol{z}^n \in E_1 \cap E_2]$$

$$(b) \quad \geq 1 - \delta/2 - \delta/2,$$

where inequality (a) is due to the relationship between $E_1 \cap E_2$ and $R(\hat{h}) \leq \min_{h \in \mathcal{H}} R(h) + \epsilon$; inequality (b) is due to the probability of union of sets.

Based on Proposition B.1, in order to guarantee $\mathbb{P}_{\boldsymbol{z}^n}\left[R_n(\boldsymbol{z}^n, \hat{h}) - E_{\boldsymbol{z}^n}[R_n(\boldsymbol{z}^n, \hat{h})] \leq \epsilon/2\right] \geq 1 - \delta/2$, $\frac{2C_r^2}{\epsilon^2} \ln \frac{4}{\delta}$ instances are required. Meanwhile, based on the definition of generalization PAC bound (Definition 5 of the main text), in order to guarantee $\mathbb{P}_{\boldsymbol{z}^n}[R(\hat{h}) - R_n(\boldsymbol{z}^n, \hat{h}) \leq \epsilon/2] \geq 1 - \delta/2$, $n_{\mathcal{H}}^G(\epsilon/2, \delta/2)$ instances are required. Therefore, with more than $\max\left[n_{\mathcal{H}}^G(\epsilon/2, \delta/2), \frac{2C_r^2}{\epsilon^2} \ln \frac{4}{\delta}\right]$ instances, $\mathbb{P}_{\boldsymbol{z}^n}\left[R(\hat{h}) \leq \min_{h \in \mathcal{H}} R(h) + \epsilon\right] \geq 1 - \delta$ is satisfied. Based on the definition of the agnostic PAC learnability (Definition 4 of the main text), the hypothesis class is agnostic PAC learnable and the agnostic PAC learner for $\mathcal{H}$ is $\mathrm{ERM}_{\mathcal{H}}$. $\square$

## B.4 Proof of Theorem 2

*Proof.* Let $E_1$ denote the set of events $R(\hat{\boldsymbol{w}}) - R_n(\boldsymbol{z}^n, \hat{\boldsymbol{w}}) \leq \epsilon$, $E_2$ denote the set of events $\hat{\boldsymbol{w}} \in \hat{\mathcal{W}}$, and $E_3$ denote the set of events $\max_{\boldsymbol{w} \in \hat{\mathcal{W}}}[R(\boldsymbol{w}) - R_n(\boldsymbol{z}^n, \boldsymbol{w})] \leq \epsilon$.

$$\mathbb{P}_{\boldsymbol{z}^n}[\neg E_1]$$

$$= \mathbb{P}_{\boldsymbol{z}^n}[\neg E_1, E_2] + \mathbb{P}_{\boldsymbol{z}^n}[\neg E_1, \neg E_2]$$

$$(a) \quad \leq \mathbb{P}_{\boldsymbol{z}^n}[\neg E_1, E_2] + \delta_1$$

$$(b) \quad \leq \mathbb{P}_{\boldsymbol{z}^n}[\neg E_3] + \delta_1$$

$$= \delta_2 + \delta_1;$$

where inequality (a) is due to $\mathbb{P}_{\boldsymbol{z}^n}[\neg E_1, \neg E_2] \leq \mathbb{P}_{\boldsymbol{z}^n}[\neg E_2] = 1 - \mathbb{P}_{\boldsymbol{z}^n}[E_2] \leq \delta_1$; inequality (b) is based on the relationship between $\mathbb{1}[E_2]\mathbb{1}[\neg E_1]$ and $\mathbb{1}[E_3]$. At the points $\boldsymbol{z}^n$ that satisfy $\boldsymbol{m}(\boldsymbol{z}^n) = \hat{\boldsymbol{w}} \in \hat{\mathcal{W}}$, $\mathbb{1}[\neg E_1] = 1 \Rightarrow \mathbb{1}[\neg E_3] = 1$, thus $\mathbb{1}[E_2]\mathbb{1}[\neg E_1] \leq \mathbb{1}[\neg E_3]$ and $\mathbb{P}_{\boldsymbol{z}^n}[\neg E_1, E_2] \leq \mathbb{P}_{\boldsymbol{z}^n}[\neg E_3]$. $\qquad\square$

## B.5 Proof of Lemma 1

*Proof.* To show that $\boldsymbol{m}_{[q]}^{(T)}(\boldsymbol{z}^n; \boldsymbol{s})$ is concentrated around its expectation, we make use of the Mc-Diarmid's Inequality. First, we note that $\boldsymbol{m}_{[q]}^{(T)}(\boldsymbol{z}^n; \boldsymbol{s}) : \mathcal{Z}^n \to \mathbb{R}$ is function mapping from random variables to a real value, and $\boldsymbol{z}^n$ satisfies the independent assumption. Second, we show that $|\boldsymbol{m}_{[q]}^{(T)}(\boldsymbol{z}^n; \boldsymbol{s}) - \boldsymbol{m}_{[q]}^{(T)}(\boldsymbol{z}^{n,i}; \boldsymbol{s})|$ is bounded. $\boldsymbol{m}^{(T)}(\boldsymbol{z}^n; \boldsymbol{s})$ and $\boldsymbol{m}_{[q]}^{(T)}(\boldsymbol{z}^n; \boldsymbol{s})$ are temporarily simplified to $\boldsymbol{m}^{(T)}(\boldsymbol{z}^n)$ and $\boldsymbol{m}_{[q]}^{(T)}(\boldsymbol{z}^n)$, respectively. $\forall \boldsymbol{s}, \forall q, |\boldsymbol{m}_{[q]}^{(T)}(\boldsymbol{z}^n) - \boldsymbol{m}_{[q]}^{(T)}(\boldsymbol{z}^{n,i})| \leq \|\boldsymbol{m}^{(T)}(\boldsymbol{z}^n) - \boldsymbol{m}^{(T)}(\boldsymbol{z}^{n,i})\|$, where $\|\cdot\|$ denotes the vector $L_2$-norm[1]. We will now discuss the bound of $\|\boldsymbol{m}^{(T)}(\boldsymbol{z}^n) - \boldsymbol{m}^{(T)}(\boldsymbol{z}^{n,i})\|$.

(1) Decompose $\boldsymbol{m}^{(t)}(\boldsymbol{z}^n)$. To understand the influence of $\boldsymbol{z}_i$, the updating equation of $\boldsymbol{m}^{(t)}(\boldsymbol{z}^n)$ is divided into two parts:

$$\boldsymbol{m}^{(t)}(\boldsymbol{z}^n) = \left(\boldsymbol{m}^{(t-1)}(\boldsymbol{z}^n) - \sum_{j \in [n]/i} \frac{\alpha^{(t)}}{n} \frac{\partial r(\boldsymbol{z}_j, \boldsymbol{w})}{\partial \boldsymbol{w}}\Big|_{\boldsymbol{m}^{(t-1)}(\boldsymbol{z}^n)}\right) - \frac{\alpha^{(t)}}{n} \frac{\partial r(\boldsymbol{z}_i, \boldsymbol{w})}{\partial \boldsymbol{w}}\Big|_{\boldsymbol{m}^{(t-1)}(\boldsymbol{z}^n)}.$$

Let $G_i(\boldsymbol{m}^{(t-1)}(\boldsymbol{z}^n); \boldsymbol{z}^n) = \boldsymbol{m}^{(t-1)}(\boldsymbol{z}^n) - \frac{\alpha^{(t)}}{n} \sum_{j \in [n]/i} \frac{\partial r(\boldsymbol{z}_j, \boldsymbol{w})}{\partial \boldsymbol{w}}\Big|_{\boldsymbol{m}^{(t-1)}(\boldsymbol{z}^n)}$. Since $G_i$ has the same formula for $\boldsymbol{z}^n$ and $\boldsymbol{z}^{n,i}$ once $\boldsymbol{m}^{(t-1)}$ is given, i.e. $G_i(\boldsymbol{m}^{(t-1)}(\boldsymbol{z}^n); \boldsymbol{z}^n) = G_i(\boldsymbol{m}^{(t-1)}(\boldsymbol{z}^n); \boldsymbol{z}^{n,i})$, the dependency of $G_i$ on the second argument can be dropped. Representing the above updating equation via $G_i$ gives:

$$\boldsymbol{m}^{(t)}(\boldsymbol{z}^n) = G_i(\boldsymbol{m}^{(t-1)}(\boldsymbol{z}^n)) - \frac{\alpha^{(t)}}{n} \frac{\partial r(\boldsymbol{z}_i, \boldsymbol{w})}{\partial \boldsymbol{w}}\Big|_{\boldsymbol{m}^{(t-1)}(\boldsymbol{z}^n)}.$$

Then,

$$\|\boldsymbol{m}^{(t)}(\boldsymbol{z}^n) - \boldsymbol{m}^{(t)}(\boldsymbol{z}^{n,i})\|$$
$$= \left\|G_i(\boldsymbol{m}^{(t-1)}(\boldsymbol{z}^n)) - \frac{\alpha^{(t)}}{n} \frac{\partial r(\boldsymbol{z}_i, \boldsymbol{w})}{\partial \boldsymbol{w}}\Big|_{\boldsymbol{m}^{(t-1)}(\boldsymbol{z}^n)} \right.$$
$$\left. - G_i(\boldsymbol{m}^{(t-1)}(\boldsymbol{z}^{n,i})) + \frac{\alpha^{(t)}}{n} \frac{\partial r(\boldsymbol{z}_i', \boldsymbol{w})}{\partial \boldsymbol{w}}\Big|_{\boldsymbol{m}^{(t-1)}(\boldsymbol{z}^{n,i})}\right\|$$
$$\leq \left\|\frac{\alpha^{(t)}}{n} \frac{\partial r(\boldsymbol{z}_i, \boldsymbol{w})}{\partial \boldsymbol{w}}\Big|_{\boldsymbol{m}^{(t-1)}(\boldsymbol{z}^n)} - \frac{\alpha^{(t)}}{n} \frac{\partial r(\boldsymbol{z}_i', \boldsymbol{w})}{\partial \boldsymbol{w}}\Big|_{\boldsymbol{m}^{(t-1)}(\boldsymbol{z}^{n,i})}\right\| \text{ (Term 1)}$$
$$+ \|G_i(\boldsymbol{m}^{(t-1)}(\boldsymbol{z}^n)) - G_i(\boldsymbol{m}^{(t-1)}(\boldsymbol{z}^{n,i}))\| \text{ (Term 2)}.$$

Term 1 and Term 2 in the inequality can be bounded by using the Lipschitz constant of a function $r$ with respect to $\boldsymbol{w}$ and the Lipschitz constant of $G$ with respect to $\boldsymbol{w}$, respectively.

(2) Bound Term 1. Recall that the Lipschitz constant is defined as:

$$\text{lip}(r \leftarrow \boldsymbol{w}) = \max_{\boldsymbol{w}_1, \boldsymbol{w}_2 \in \mathcal{W}, \boldsymbol{w}_1 \neq \boldsymbol{w}_2, \boldsymbol{z} \in \mathcal{Z}} \frac{|r(\boldsymbol{z}; \boldsymbol{w}_1) - r(\boldsymbol{z}; \boldsymbol{w}_2)|}{\|\boldsymbol{w}_1 - \boldsymbol{w}_2\|}.$$

Term 1 is bounded as follows:

$$\left\|\frac{\alpha^{(t)}}{n} \frac{\partial r(\boldsymbol{z}_i, \boldsymbol{w})}{\partial \boldsymbol{w}}\Big|_{\boldsymbol{m}^{(t-1)}(\boldsymbol{z}^n)} - \frac{\alpha^{(t)}}{n} \frac{\partial r(\boldsymbol{z}_i', \boldsymbol{w})}{\partial \boldsymbol{w}}\Big|_{\boldsymbol{m}^{(t-1)}(\boldsymbol{z}^{n,i})}\right\|$$
$$\leq \left\|\frac{\alpha^{(t)}}{n} \frac{\partial r(\boldsymbol{z}_i, \boldsymbol{w})}{\partial \boldsymbol{w}}\Big|_{\boldsymbol{m}^{(t-1)}(\boldsymbol{z}^n)}\right\| + \left\|\frac{\alpha^{(t)}}{n} \frac{\partial r(\boldsymbol{z}_i', \boldsymbol{w})}{\partial \boldsymbol{w}}\Big|_{\boldsymbol{m}^{(t-1)}(\boldsymbol{z}^{n,i})}\right\|$$
$$\leq \frac{2\alpha^{(t)}}{n} \text{lip}(r \leftarrow \boldsymbol{w}).$$

(3) Bound Term 2. Let $\eta_i = \mathrm{lip}\,(G_i \leftarrow \boldsymbol{w})$.

$$\|G_i(\boldsymbol{m}^{(t-1)}(\boldsymbol{z}^n)) - G_i(\boldsymbol{m}^{(t-1)}(\boldsymbol{z}^{n,i}))\| \le \eta_i \|\boldsymbol{m}^{(t-1)}(\boldsymbol{z}^n) - \boldsymbol{m}^{(t-1)}(\boldsymbol{z}^{n,i})\|$$

(4) Bound $\|\boldsymbol{m}^{(T)}(\boldsymbol{z}^n) - \boldsymbol{m}^{(T)}(\boldsymbol{z}^{n,i})\|$ and $|\boldsymbol{m}_{[q]}^{(T)}(\boldsymbol{z}^n) - \boldsymbol{m}_{[q]}^{(T)}(\boldsymbol{z}^{n,i})|$.

$t = 1$

$$\|\boldsymbol{m}^{(1)}(\boldsymbol{z}^n) - \boldsymbol{m}^{(1)}(\boldsymbol{z}^{n,i})\|$$
$$\le \left\| \frac{\alpha^{(1)}}{n} \frac{\partial r(\boldsymbol{z}_i, \boldsymbol{w})}{\partial \boldsymbol{w}}|_{\boldsymbol{w}^0} - \frac{\alpha^{(1)}}{n} \frac{\partial r(\boldsymbol{z}_i', \boldsymbol{w})}{\partial \boldsymbol{w}}|_{\boldsymbol{w}^0} \right\| + \|G_i(\boldsymbol{w}^0) - G_i(\boldsymbol{w}^0)\|$$
$$\le \frac{2\alpha^{(1)}}{n} \mathrm{lip}\,(r \leftarrow \boldsymbol{w});$$

$t = 2$

$$\|\boldsymbol{m}^{(2)}(\boldsymbol{z}^n) - \boldsymbol{m}^{(2)}(\boldsymbol{z}^{n,i})\|$$
$$\le \left\| \frac{\alpha^{(2)}}{n} \frac{\partial r(\boldsymbol{z}_i, \boldsymbol{w})}{\partial \boldsymbol{w}}|_{\boldsymbol{m}^{(1)}(\boldsymbol{z}^n)} - \frac{\alpha^{(2)}}{n} \frac{\partial r(\boldsymbol{z}_i', \boldsymbol{w})}{\partial \boldsymbol{w}}|_{\boldsymbol{m}^{(1)}(\boldsymbol{z}^{n,i})} \right\|$$
$$+ \|G_i(\boldsymbol{m}^{(1)}(\boldsymbol{z}^n)) - G_i(\boldsymbol{m}^{(1)}(\boldsymbol{z}^{n,i}))\|$$
$$\le \frac{2\alpha^{(2)}}{n} \mathrm{lip}\,(r \leftarrow \boldsymbol{w}) + \eta_i \frac{2\alpha^{(1)}}{n} \mathrm{lip}\,(r \leftarrow \boldsymbol{w})$$
$$= \frac{2(\eta_i \alpha^{(1)} + \alpha^{(2)}) \mathrm{lip}\,(r \leftarrow \boldsymbol{w})}{n};$$

$\vdots$

$t = T$

$$\|\boldsymbol{m}^{(T)}(\boldsymbol{z}^n) - \boldsymbol{m}^{(T)}(\boldsymbol{z}^{n,i})\|$$
$$\le \left\| \frac{\alpha^{(T)}}{n} \frac{\partial r(\boldsymbol{z}_i, \boldsymbol{w})}{\partial \boldsymbol{w}}|_{\boldsymbol{m}^{(T-1)}(\boldsymbol{z}^n)} - \frac{\alpha^{(T)}}{n} \frac{\partial r(\boldsymbol{z}_i', \boldsymbol{w})}{\partial \boldsymbol{w}}|_{\boldsymbol{m}^{(T-1)}(\boldsymbol{z}^{n,i})} \right\|$$
$$+ \|G_i(\boldsymbol{m}^{(T-1)}(\boldsymbol{z}^n)) - G_i(\boldsymbol{m}^{(T-1)}(\boldsymbol{z}^{n,i}))\|$$
$$\le \frac{2\left(\sum_{t=1}^{T} \eta_i^{T-t} \alpha^{(t)}\right) \mathrm{lip}\,(r \leftarrow \boldsymbol{w})}{n}.$$

Let $C_i = 2\left(\sum_{t=1}^{T} \eta_i^{T-t} \alpha^{(t)}\right) \mathrm{lip}\,(r \leftarrow \boldsymbol{w})$. Then, $|\boldsymbol{m}_{[q]}^{(T)}(\boldsymbol{z}^n) - \boldsymbol{m}_{[q]}^{(T)}(\boldsymbol{z}^{n,i})| \le \frac{C_i}{n}$.

(5) Derive the concentration inequality. Based on Lemma B.1, $\boldsymbol{m}_{[q]}^{(T)}(\boldsymbol{z}^n)$ can be bounded as

$$\mathbb{P}_{\boldsymbol{z}^n}\left[ \left| \boldsymbol{m}_{[q]}^{(T)}(\boldsymbol{z}^n) - \mathbb{E}_{\boldsymbol{z}^n} \boldsymbol{m}_{[q]}^{(T)}(\boldsymbol{z}^n) \right| \le \frac{\epsilon}{\sqrt{Q}} \right] \ge 1 - 2\exp\left( \frac{-2\epsilon^2}{Q \sum_{i=1}^{n}(C_i/n)^2} \right)$$
$$\ge 1 - 2\exp\left( \frac{-2\epsilon^2 n}{QC^2} \right),$$

where $C = 2\left(\sum_{t=1}^{T} \eta^{T-t} \alpha^{(t)}\right) \mathrm{lip}\,(r \leftarrow \boldsymbol{w})$ and $\eta = \max_{i \in [n]} \mathrm{lip}\,(G_i \leftarrow \boldsymbol{w})$.

Therefore,

$$\mathbb{P}_{\boldsymbol{z}^n}[\|\boldsymbol{m}^{(T)}(\boldsymbol{z}^n) - \mathbb{E}_{\boldsymbol{z}^n} \boldsymbol{m}^{(T)}(\boldsymbol{z}^n)\| \le \epsilon]$$
$$(a) \quad \ge \mathbb{P}_{\boldsymbol{z}^n}\left[ \bigcap_{q=1}^{Q} |\boldsymbol{m}_{[q]}^{(T)}(\boldsymbol{z}^n) - \mathbb{E}_{\boldsymbol{z}^n} \boldsymbol{m}_{[q]}^{(T)}(\boldsymbol{z}^n)| \le \frac{\epsilon}{\sqrt{Q}} \right]$$
$$(b) \quad \ge 1 - 2Q\exp\left( \frac{-2\epsilon^2 n}{QC^2} \right),$$

where inequality (a) is due the relationship between the events; inequality (b) is due to a Frechet inequality. $\square$

## B.6 Proof of Lemma 2

First, the definitions of Rademacher complexity, uniform convergence and covering number are introduced. Dudley's Integral Theorem that uses covering number to bound Rademacher complexity is also introduced. Then, by using the Lipschitz constant, the covering number of functional space is shown to be bounded by the covering number of parameter space. Finally, based on Dudley's Integral Theorem, Lemma 2 is shown.

### B.6.1 Preliminary

**Definition B.1.** [6] Let $\epsilon^n = \{\epsilon_1, \dots \epsilon_n\}$ be i.i.d. random variables with $P(\epsilon_i = 1) = P(\epsilon_i = -1) = \frac{1}{2}$. $z^n = \{z_1, \dots, z_n\}$ are i.i.d. samples. The *empirical Rademacher complexity* is defined as

$$\hat{\mathrm{Rad}}_n(\mathcal{H}) = \mathbb{E}_{\epsilon^n}\Big[\max_{h \in \mathcal{H}} \frac{1}{n}\sum_i \epsilon_i h(z_i)\Big|z^n\Big];$$

and the *Rademacher complexity* is defined as

$$\mathrm{Rad}(\mathcal{H}) = \mathbb{E}_{z^n}\Big[\hat{\mathrm{Rad}}_n(\mathcal{H})\Big].$$

**Theorem B.1.** [6] With probability at least $1 - \delta$ the following bound holds:

$$R(h) - R_n(z^n, h) \leq 2\hat{\mathrm{Rad}}_n(\phi \circ \mathcal{H}) + 3\sqrt{\frac{\ln\frac{2}{\delta}}{2n}},$$

where $\phi : \mathbb{R} \to \mathbb{R}$ denotes the loss function $l(h(x); y)$; $\circ$ denotes the composition of functions.

**Lemma B.3.** [6] Let $\phi : \mathbb{R} \to \mathbb{R}$ be an $L$-Lipschitz function. Then, for any hypothesis set $\mathcal{H}$ of real-valued functions, *Talagrand's Lemma* indicates the following inequality holds:

$$\hat{\mathrm{Rad}}_n(\phi \circ \mathcal{H}) \leq L\hat{\mathrm{Rad}}_n(\mathcal{H}).$$

**Corollary B.1.** Suppose $\mathrm{lip}(r \leftarrow h) \leq L$, then with probability at least $1 - \delta$ the following bound holds:

$$R(h) - R_n(z^n, h) \leq 2L\hat{\mathrm{Rad}}_n(\mathcal{H}) + 3\sqrt{\frac{\ln\frac{2}{\delta}}{2n}}$$

*Proof.* Substituting the result of Lemma B.3 into Theorem B.1 gives the result. □

**Definition B.2.** [8] An $\epsilon$-*cover* of a subset $\mathcal{U}$ of a metric space $(\mathcal{V}, \rho)$ is a set $\hat{\mathcal{U}} \subseteq \mathcal{U}$ such that for each $u \in \mathcal{U}$ there is a $\hat{u} \in \hat{\mathcal{U}}$ such that $\rho(u, \hat{u}) \leq \epsilon$. The $\epsilon$-*cover number* of $\mathcal{U}$ is

$$N(\epsilon, \mathcal{U}, \rho) = \min\{|\hat{\mathcal{U}}| : \hat{\mathcal{U}} \text{ is an } \epsilon\text{-cover of } \mathcal{U}\}.$$

The following theorem illustrates how to bound the covering number.

**Theorem B.2.** [8] Let $\mathcal{U} \subseteq \mathcal{V} = \mathbb{R}^D$. Then

$$\Big(\frac{1}{\epsilon}\Big)^D \frac{\mathrm{vol}(\mathcal{U})}{\mathrm{vol}(\mathcal{B})} \leq N(\epsilon, \mathcal{U}, \|\cdot\|) \leq \Big(\frac{\mathrm{vol}(\mathcal{U} + \frac{\epsilon}{2}\mathcal{B})}{\mathrm{vol}(\frac{\epsilon}{2}\mathcal{B})}\Big)$$

where $+$ is the Minkovski sum, $\mathcal{B}$ is the unit norm ball and vol indicates the volume of the set.

Remark: Consider $\mathcal{U} \in \mathbb{R}^D$ with diameter $\mathrm{diam}(\mathcal{U})$. Based on the last inequality, we have

$$N(\epsilon, \mathcal{U}, \|\cdot\|) \leq \Big(\frac{\mathrm{vol}(\mathcal{U} + \frac{\epsilon}{2}\mathcal{B})}{\mathrm{vol}(\frac{\epsilon}{2}\mathcal{B})}\Big) \leq \Big(\frac{\mathrm{diam}(\mathcal{U}) + \epsilon}{\epsilon}\Big)^D = \Big(1 + \frac{\mathrm{diam}(\mathcal{U})}{\epsilon}\Big)^D.$$

**Definition B.3.** Let $\forall h_1, h_2 \in \mathcal{H}$ be two functions mapping $z \in \mathcal{Z}$ into real value, $\rho_{\mathcal{H}|z^n}$ is defined as follows:

$$\rho_{\mathcal{H}|z^n}(h_1, h_2) = \sqrt{\frac{1}{n}\sum_{i=1}^n (h_1(z_i) - h_2(z_i))^2}.$$

**Theorem B.3.** [7] With metric $\rho_{\mathcal{H}|\boldsymbol{z}^n}$ on $\mathcal{H}$, *Dudley's integral* indicates

$$\hat{\text{Rad}}_n(\mathcal{H}) \le 12 \int_0^\infty \sqrt{\frac{\log N(\epsilon, \mathcal{H}, \rho_{\mathcal{H}|\boldsymbol{z}^n})}{n}} d\epsilon.$$

Dudley's integral bounds the empirical Rademacher complexity by the covering number of the function space (with a metric based on the difference of the function value on $n$ inputs).

### B.6.2 Bound of the covering number of functional space

To start with, another definition of metric in function space is given as follows.

**Definition B.4.** A metric $\rho_{\mathcal{H}_{\boldsymbol{w}}}$ in parametric function space is defined as follows:

$$\rho_{\mathcal{H}_{\boldsymbol{w}}}(h(\cdot; \boldsymbol{w}_1), h(\cdot; \boldsymbol{w}_2)) = \max_{\boldsymbol{x} \in \mathcal{X}} |h(\boldsymbol{x}; \boldsymbol{w}_1) - h(\boldsymbol{x}; \boldsymbol{w}_2)|. \tag{4}$$

$\text{lip}(h; \mathcal{H}_{\boldsymbol{w}} \leftarrow \mathcal{W})$ will be written as $\text{lip}(h \leftarrow \boldsymbol{w})$ if $\mathcal{W}$ and $\mathcal{H}_{\boldsymbol{w}}$ are clear from the context:

$$\text{lip}(h \leftarrow \boldsymbol{w}) = \max_{\boldsymbol{w}_1, \boldsymbol{w}_2 \in \mathcal{W}, \boldsymbol{w}_1 \ne \boldsymbol{w}_2} \frac{\rho_{\mathcal{H}_{\boldsymbol{w}}}\left(h(\cdot; \cdot, \boldsymbol{w}_1), h(\cdot; \cdot, \boldsymbol{w}_2)\right)}{\rho_{\mathcal{W}}(\boldsymbol{w}_1, \boldsymbol{w}_2)}$$

$$= \max_{\boldsymbol{w}_1, \boldsymbol{w}_2 \in \mathcal{W}, \boldsymbol{w}_1 \ne \boldsymbol{w}_2, \boldsymbol{x}} \frac{|h(\boldsymbol{x}; \boldsymbol{w}_1) - h(\boldsymbol{x}; \boldsymbol{w}_2)|}{\|\boldsymbol{w}_1 - \boldsymbol{w}_2\|}.$$

**Proposition B.2.** For all spaces of parametric functions $\mathcal{H}_{\boldsymbol{w}}, \forall \epsilon, \forall \mathcal{H}$,

$$N(\epsilon, \mathcal{H}, \rho_{\mathcal{H}|\boldsymbol{z}^n}) \le N(\epsilon, \mathcal{H}, \rho_{\mathcal{H}_{\boldsymbol{w}}}), \tag{5}$$

where $\boldsymbol{w}$ denotes all parameters of the function, $\rho_{\mathcal{H}|\boldsymbol{z}^n}$ is defined in Definition B.3 and $\rho_{\mathcal{H}_{\boldsymbol{w}}}$ is defined in Definition B.4.

*Proof.* Let $\{\hat{h}_1, \ldots, \hat{h}_N\}$ be an $\epsilon$-covering set in $\mathcal{H}_{\boldsymbol{w}}$ with metric $\rho_{\mathcal{H}_{\boldsymbol{w}}}$, then based on the definition of covering set,

$$\forall h \in \mathcal{H}, \min_j \rho_{\mathcal{H}_{\boldsymbol{w}}}(h, \hat{h}_j) \le \epsilon.$$

Based on the definitions of $\rho_{\mathcal{H}|\boldsymbol{z}^n}$ and $\rho_{\mathcal{H}_{\boldsymbol{w}}}$, we have

$$\rho_{\mathcal{H}|\boldsymbol{z}^n}(h, \hat{h}_j) = \sqrt{\frac{1}{n} \sum_{i=1}^n (h(\boldsymbol{z}_i) - \hat{h}_j(\boldsymbol{z}_i))^2} \le \sqrt{\frac{1}{n} \sum_{i=1}^n \left(\max_{\boldsymbol{z}} |h(\boldsymbol{z}) - \hat{h}_j(\boldsymbol{z})|\right)^2}$$

$$= \sqrt{\frac{1}{n} \times n \times \left(\rho_{\mathcal{H}_{\boldsymbol{w}}}(h, \hat{h}_j)\right)^2} = \rho_{\mathcal{H}_{\boldsymbol{w}}}(h, \hat{h}_j) \le \epsilon.$$

Therefore, $\{\hat{h}_1, \ldots, \hat{h}_N\}$ is also an $\epsilon$-covering set of $\mathcal{H}_{\boldsymbol{w}}$ with metric $\rho_{\mathcal{H}|\boldsymbol{z}^n}$ and

$$N(\epsilon, \mathcal{H}, \rho_{\mathcal{H}|\boldsymbol{z}^n}) \le |\{\hat{h}_1, \ldots, \hat{h}_N\}| = N(\epsilon, \mathcal{H}, \rho_{\mathcal{H}_{\boldsymbol{w}}}).$$

$\square$

**Corollary B.2.** The empirical Rademacher complexity can be bounded by the covering number with metric $\rho_{\mathcal{H}_{\boldsymbol{w}}}$ as follows:

$$\hat{\text{Rad}}_n(\mathcal{H}) \le 12 \int_0^\infty \sqrt{\frac{\log N(\epsilon, \mathcal{H}, \rho_{\mathcal{H}_{\boldsymbol{w}}})}{n}} d\epsilon.$$

*Proof.* Substituting the result of Proposition B.2 into Theorem B.3 gives the result. $\square$

**Proposition B.3.** Let $h(\boldsymbol{z}; \boldsymbol{w})$ be a parameterized function and $\boldsymbol{w} \in \mathcal{W} \in \mathbb{R}^Q$. Suppose $\text{lip}(h \leftarrow \boldsymbol{w}) \le L$. Then,

$$N(\epsilon, \mathcal{H}_{\boldsymbol{w}}, \rho_{\mathcal{H}_{\boldsymbol{w}}}) \le N(\epsilon/L, \mathcal{W}, \rho_{\mathcal{W}}) \le \left(1 + \frac{\text{diam}(\mathcal{W})L}{\epsilon}\right)^Q.$$

*Proof.* The second inequality follows from Theorem B.2. We now show the first inequality. Let $\{\hat{\boldsymbol{w}}_1, \ldots, \hat{\boldsymbol{w}}_N\}$ be an $(\epsilon/L)$-covering set in $\mathcal{W}$. Based on the definition of covering set,

$$\forall \boldsymbol{w} \in \mathcal{W}, \min_i \rho_{\mathcal{W}}(\boldsymbol{w}, \hat{\boldsymbol{w}}_i) \leq \epsilon/L.$$

Based on the definition of Lipschitz constant,

$$\forall h(\cdot; \boldsymbol{w}) \in \mathcal{H}_{\boldsymbol{w}}, \min_i \rho_{\mathcal{H}_{\boldsymbol{w}}}\Big(h(\cdot; \boldsymbol{w}), h(\cdot; \hat{\boldsymbol{w}}_i)\Big) \leq L \min_i \rho_{\mathcal{W}}(\boldsymbol{w}, \hat{\boldsymbol{w}}_i) \leq \epsilon.$$

Therefore, $\{h(\cdot; \hat{\boldsymbol{w}}_1), \ldots, h(\cdot; \hat{\boldsymbol{w}}_N)\}$ is a $\epsilon$-covering set of $\mathcal{H}$ and

$$N(\epsilon, \mathcal{H}(\boldsymbol{w}), \rho_{\mathcal{H}_{\boldsymbol{w}}}) \overset{(c)}{\leq} |\{h(\cdot; \hat{\boldsymbol{w}}_1), \ldots, h(\cdot; \hat{\boldsymbol{w}}_N)\}| \overset{(d)}{\leq} |\{\hat{\boldsymbol{w}}_1, \ldots, \hat{\boldsymbol{w}}_N\}| = N(\epsilon/L, \mathcal{W}, \rho_{\mathcal{W}}),$$

where inequality (c) is based on the definition of covering number; inequality (d) is due to the fact that $h$ is a function. □

### B.6.3  Proof of Lemma 2

*Proof.* Based on the result of Corollary B.2,

$$\hat{\mathrm{Rad}}_n(\mathcal{H})$$

$$\leq 12 \int_0^\infty \sqrt{\frac{\log N(\epsilon, \mathcal{H}, \rho_{\mathcal{H}_w})}{n}} d\epsilon \overset{(a)}{=} 12 \int_0^{LB} \sqrt{\frac{\log N(\epsilon, \mathcal{H}, \rho_{\mathcal{H}_w})}{n}} d\epsilon$$

$$\overset{(b)}{\leq} \frac{12}{\sqrt{n}} \int_0^{LB} \sqrt{\log\left(1 + \frac{LB}{\epsilon}\right)^Q} d\epsilon \overset{(c)}{=} \frac{12LB}{\sqrt{n}} \int_0^1 \sqrt{Q \log\left(1 + \frac{1}{\epsilon'}\right)} d\epsilon'$$

$$\overset{(d)}{\leq} 12LB\sqrt{\frac{Q}{n}} \int_0^1 \sqrt{\log(2/\epsilon')} d\epsilon' \overset{(e)}{=} 24LB\sqrt{\frac{Q}{n}} \int_0^{1/2} \sqrt{\log(1/\epsilon)} d\epsilon.$$

Equality (a) holds as the value of $h$ is bounded by $LB$; if $\epsilon > LB$, then $\log N(\epsilon, \mathcal{H}, \rho_{\mathcal{H}_w}) = 0$; inequality (b) is based on Proposition B.3; equality (c) follows from variable substitution $\epsilon' = \frac{\epsilon}{LB}$; inequality (d) is due to $\epsilon' \in [0, 1]$; equality (e) follows from another variable substitution $\epsilon = \frac{\epsilon'}{2}$.

Then we calculate the integral

$$\int_0^{1/2} \sqrt{\log(1/\epsilon)} d\epsilon$$

$$\overset{(a)}{=} \int_\infty^{\sqrt{log2}} y d(e^{-y^2}) \overset{(b)}{=} e^{-y^2} y|_\infty^{\sqrt{\log 2}} - \int_\infty^{\sqrt{\log 2}} e^{-y^2} dy$$

$$= e^{-y^2} y|_\infty^{\sqrt{\log 2}} + \int_{\sqrt{\log 2}}^\infty e^{-y^2} dy \leq e^{-y^2} y|_\infty^{\sqrt{\log 2}} + \int_0^\infty e^{-y^2} dy$$

$$= \frac{\sqrt{\log 2}}{2} + \frac{\sqrt{\pi}}{2},$$

where equality (a) is based on variable substitution $y = \sqrt{\log(1/\epsilon)}$, i.e. $\epsilon = e^{-y^2}$ and equality (b) is based on integration by parts.

Therefore,

$$\hat{\mathrm{Rad}}_n(\mathcal{H}) \leq 24LB\sqrt{\frac{Q}{n}} \int_0^{1/2} \sqrt{\log(1/\epsilon)} d\epsilon$$

$$\leq 24\left(\frac{\sqrt{\log 2}}{2} + \frac{\sqrt{\pi}}{2}\right) LB\sqrt{\frac{Q}{n}}$$

$$= CLB\sqrt{\frac{Q}{n}},$$

where $C = 12(\sqrt{\log 2} + \sqrt{\pi})$.
Finally, substituting the above bound of empirical Rademacher complexity into Corollary B.1 gives Lemma 2. □

## B.7 Proof of Theorem 3

*Proof.* Let $\mathrm{ball}(E, \epsilon) := \mathrm{ball}\left(\mathbb{E}_{\boldsymbol{z}^n} \boldsymbol{m}^{(T)}(\boldsymbol{z}^n), \epsilon\right)$ denote the ball with the center at $\mathbb{E}_{\boldsymbol{z}^n} \boldsymbol{m}^{(T)}(\boldsymbol{z}^n)$ and radius of $\epsilon$. Let $L = \mathrm{lip}(r \leftarrow \boldsymbol{w})$. Based on Lemma 1, we have

$$\mathbb{P}_{\boldsymbol{z}^n}[\boldsymbol{m}(\boldsymbol{z}^n) \in \mathrm{ball}(E, \epsilon)] \geq 1 - \delta_1, \tag{6}$$

where $\delta_1 = 2Q \exp(\frac{-2\epsilon^2 n}{Q(2C_2)^2 L^2})$, that is $\epsilon = C_2 L \sqrt{\frac{2Q}{n} \ln \frac{2Q}{\delta_1}}$.

Based on the result of Lemma 2,

$$\mathbb{P}_{\boldsymbol{z}^n}\left[\max_{\boldsymbol{w} \in \mathrm{ball}(E, \epsilon)} \left(R(\boldsymbol{w}) - R_n(\boldsymbol{z}^n, \boldsymbol{w})\right) \leq CL(2\epsilon)\sqrt{\frac{Q}{n}} + \sqrt{\frac{\ln 1/\delta_2}{2n}}\right] \geq 1 - \delta_2.$$

Substituting $\epsilon = C_2 L \sqrt{\frac{2Q}{n} \ln \frac{2Q}{\delta_1}} \leq C_2 L_1 L_l \sqrt{\frac{2Q}{n} \ln \frac{2Q}{\delta_1}}$ into the above formula, we have

$$\mathbb{P}_{\boldsymbol{z}^n}\left[\max_{\boldsymbol{w} \in \mathrm{ball}(E, \epsilon)} \left(R(\boldsymbol{w}) - R_n(\boldsymbol{z}^n, \boldsymbol{w})\right) \leq \epsilon'\right] \geq 1 - \delta_2, \tag{7}$$

where

$$\epsilon' = \frac{2CC_2 L_1^2 L_l^2 Q \sqrt{2 \ln(2Q/\delta_1)}}{n} + \sqrt{\frac{\ln(1/\delta_2)}{2n}}.$$

Based on Theorem 2, the final result is obtained by combining Eqs. 6,7 and setting $C_1 = 2\sqrt{2}C$:

$$\mathbb{P}_{\boldsymbol{z}^n}[R(\boldsymbol{m}(\boldsymbol{z}^n)) - R_n(\boldsymbol{z}^n, \boldsymbol{m}(\boldsymbol{z}^n)) \leq \epsilon] \geq 1 - \delta_1 - \delta_2.$$

$\square$

# C  Lipschitz smoothness and updating equations of SMILE

For a classifier $h$ with convex constraints on parameters, the parameter $\boldsymbol{w}$ will be restricted to be inside a convex set, as explained in Sec. C.1. Then based on Corollary A.1, a sufficient condition for bounded $\mathrm{lip}(\frac{\partial h}{\partial \boldsymbol{w}} \leftarrow \boldsymbol{w})$ is to have finite values of the first and second partial derivatives. Conditions when SMILE satisfies these two conditions are shown in Secs. C.2 and C.3, respectively. Based on these results, the generalization bound of SMILE is proved in Sec. C.4 and its updating equations are given in Sec. C.5.

## C.1  Equivalence between constrained optimization and the use of regularization terms

Let us review two optimization problems.
Problem 1:

$$\min_{\boldsymbol{w}} R_n(\boldsymbol{z}_n, h_{\boldsymbol{w}}) \quad \text{s.t. } \mathcal{P}(\boldsymbol{w}) \leq C;$$

Problem 2:

$$\min_{\boldsymbol{w}} \quad R_n(\boldsymbol{z}_n, h_{\boldsymbol{w}}) + \lambda \mathcal{P}(\boldsymbol{w}).$$

The Lagrange function of Problem 1 is

$$\mathcal{L}(\boldsymbol{w}, u) = R_n(\boldsymbol{z}_n, h_{\boldsymbol{w}}) + u(\mathcal{P}(\boldsymbol{w}) - C), \quad u \geq 0,$$

where $u$ is the Lagrangian multiplier.
For Problem 1, the (KKT) necessary conditions imply

$$\text{Condition 1} \quad \frac{\partial R_n(\boldsymbol{z}_n, h_{\boldsymbol{w}})}{\partial \boldsymbol{w}} + u \frac{\partial \mathcal{P}(\boldsymbol{w})}{\partial \boldsymbol{w}} = 0;$$

$$\text{Condition 2} \quad u(\mathcal{P}(\boldsymbol{w}) - C) = 0.$$

For Problem 2, the necessary condition implies

$$\frac{\partial R_n(\boldsymbol{z}_n, h_{\boldsymbol{w}})}{\partial \boldsymbol{w}} + \lambda \frac{\partial \mathcal{P}(\boldsymbol{w})}{\partial \boldsymbol{w}} = 0.$$

Suppose $\boldsymbol{w}_1^*$ and $\mu^*$ satisfy the necessary condition of Problem 1. Setting $\lambda = \mu^*$, we can see that $\boldsymbol{w}_1^*$ satisfies for the necessary condition of Problem 2. Suppose $\boldsymbol{w}_2^*$ satisfies the necessary condition of Problem 2. Setting $\mu = \lambda$ and $C = \mathcal{P}(\boldsymbol{w}_2^*)$, we can see that Condition 1 and Condition 2 of Problem 1 are satisfied, so $\boldsymbol{w}_2^*$ satisfies the necessary condition of Problem 1 as well. Based on the above results, the necessary conditions of Problem 1 and Problem 2 are equivalent.

Meanwhile, when the regularization term in Problem 2 is a convex function, the equivalent Problem 1 constrains $\boldsymbol{w}$ inside the set of $\{\boldsymbol{w}|\mathcal{P}(\boldsymbol{w}) \leq C\}$, which is a convex set [2].

## C.2 First partial derivatives of SMILE classifier

The first partial derivatives of the classifier (Eq. 8) are as follows:

$$\frac{\partial h(\boldsymbol{x};\Theta)}{\partial \boldsymbol{r}_j^+} = -\exp(-\|\boldsymbol{L}\boldsymbol{x} - \boldsymbol{r}_j^+\|^2)(2\boldsymbol{r}_j^+ - 2\boldsymbol{L}\boldsymbol{x})$$

$$\frac{\partial h(\boldsymbol{x};\Theta)}{\partial \boldsymbol{r}_k^-} = \exp(-\|\boldsymbol{L}\boldsymbol{x} - \boldsymbol{r}_k^-\|^2)(2\boldsymbol{r}_k^- - 2\boldsymbol{L}\boldsymbol{x})$$

$$\frac{\partial h(\boldsymbol{x};\Theta)}{\partial \boldsymbol{L}_{[a,b]}} = -\sum_j 2(\boldsymbol{L}\boldsymbol{x} - \boldsymbol{r}_j^+)_{[a]}\boldsymbol{x}_{[b]}\exp(-\|\boldsymbol{L}\boldsymbol{x} - \boldsymbol{r}_j^+\|^2)$$

$$+ \sum_k 2(\boldsymbol{L}\boldsymbol{x} - \boldsymbol{r}_k^-)_{[a]}\boldsymbol{x}_{[b]}\exp(-\|\boldsymbol{L}\boldsymbol{x} - \boldsymbol{r}_k^-\|^2),$$

where $\boldsymbol{L}_{[a,b]}$ denotes the $a$th row and $b$th column element of matrix $\boldsymbol{L}$ and $\boldsymbol{x}_{[a]}$ denotes the $a$th element of the vector $\boldsymbol{x}$; $(\boldsymbol{L}\boldsymbol{x} - \boldsymbol{r})_{[a]} = \sum_i \boldsymbol{L}_{[ai]}\boldsymbol{x}_{[i]} - \boldsymbol{r}_{[a]}$.

$\frac{\partial h(\boldsymbol{x};\Theta)}{\partial \boldsymbol{r}_j^+}$ and $\frac{\partial h(\boldsymbol{x};\Theta)}{\partial \boldsymbol{r}_k^-}$ are bounded by $2\operatorname{diam}(\boldsymbol{r}) + 2\operatorname{diam}(\boldsymbol{L})\operatorname{diam}(\boldsymbol{x})$; $\frac{\partial h(\boldsymbol{x};\Theta)}{\partial \boldsymbol{L}}$ is bounded by $4m(\operatorname{diam}(\boldsymbol{r}) + \operatorname{diam}(\boldsymbol{L})\operatorname{diam}(\boldsymbol{x}))\operatorname{diam}(\boldsymbol{x})$, where $m$ denotes the number of representative instances. All first partial derivatives have finite values as long as $\operatorname{diam}(\boldsymbol{L})$, $\operatorname{diam}(\boldsymbol{x})$ and $\operatorname{diam}(\boldsymbol{r})$ are bounded.

## C.3 Second partial derivatives of SMILE classifier

The (unmixed) second partial derivatives are as follows:

$$\frac{\partial^2 h(\boldsymbol{x};\Theta)}{\partial \boldsymbol{r}_j^{+2}} = 4\exp(-\|\boldsymbol{L}\boldsymbol{x} - \boldsymbol{r}_j^+\|^2)(\boldsymbol{r}_j^+ - \boldsymbol{L}\boldsymbol{x})(\boldsymbol{r}_j^+ - \boldsymbol{L}\boldsymbol{x})^T - 2\exp(-\|\boldsymbol{L}\boldsymbol{x} - \boldsymbol{r}_j^+\|^2)\boldsymbol{I};$$

$$\frac{\partial^2 h(\boldsymbol{x};\Theta)}{\partial \boldsymbol{r}_k^{-2}} = -4\exp(-\|\boldsymbol{L}\boldsymbol{x} - \boldsymbol{r}_k^-\|^2)(\boldsymbol{r}_k^- - \boldsymbol{L}\boldsymbol{x})(\boldsymbol{r}_k^- - \boldsymbol{L}\boldsymbol{x})^T + 2\exp(-\|\boldsymbol{L}\boldsymbol{x} - \boldsymbol{r}_k^-\|^2)\boldsymbol{I};$$

$$\frac{\partial^2 h(\boldsymbol{x};\Theta)}{\partial \boldsymbol{L}_{[a,b]}^2} = \sum_j 4(\boldsymbol{L}\boldsymbol{x} - \boldsymbol{r}_j^+)_{[a]}^2 \boldsymbol{x}_{[b]}^2 \exp(-\|\boldsymbol{L}\boldsymbol{x} - \boldsymbol{r}_j^+\|^2) - 2\sum_j \boldsymbol{x}_{[b]}^2 \exp(-\|\boldsymbol{L}\boldsymbol{x} - \boldsymbol{r}_j^+\|^2)$$

$$- \sum_k 4(\boldsymbol{L}\boldsymbol{x} - \boldsymbol{r}_k^-)_{[a]}^2 \boldsymbol{x}_{[b]}^2 \exp(-\|\boldsymbol{L}\boldsymbol{x} - \boldsymbol{r}_k^-\|^2) + 2\sum_k \boldsymbol{x}_{[b]}^2 \exp(-\|\boldsymbol{L}\boldsymbol{x} - \boldsymbol{r}_k^-\|^2),$$

where $\boldsymbol{I}$ is the identity matrix. The mixed partials, e.g. $\frac{\partial^2 h}{\partial \boldsymbol{r}_j^+ \partial \boldsymbol{r}_k^-}$, $\frac{\partial^2 h}{\partial \boldsymbol{r}_j^+ \partial \boldsymbol{L}_{[a,b]}}$, can be derived similarly. All second partial derivatives have finite values as long as $\operatorname{diam}(\boldsymbol{L})$, $\operatorname{diam}(\boldsymbol{x})$ and $\operatorname{diam}(\boldsymbol{r})$ are bounded.

## C.4 Generalization bound of SMILE

**Corollary C.1.** Let $\boldsymbol{\theta} \in \Theta = \{\boldsymbol{r}^m, \boldsymbol{L}\} \in \mathbb{R}^Q$ and $\hat{\boldsymbol{\theta}}$ denote the learned parameters. Let $\mathcal{R}$ and $\mathcal{X}$ denote the set of all possible values of $\boldsymbol{r}$ and $\boldsymbol{x}$ respectively, and assume $\{\boldsymbol{L}\boldsymbol{x}\} \subseteq \mathcal{R}$. Then with probability at least $1 - \delta_1 - \delta_2$, the following bound holds:

$$R(\hat{\boldsymbol{\theta}}) - R_n(\boldsymbol{z}^n, \hat{\boldsymbol{\theta}}) \le \frac{C_1 C_2 Q \sqrt{\ln(2Q/\delta_1)}}{n} + \sqrt{\frac{\ln(1/\delta_2)}{2n}}, \tag{8}$$

where $C_1 = 96\sqrt{2}(\sqrt{\log 2} + \sqrt{\pi})m\operatorname{diam}(\mathcal{R})^2(1 + m\operatorname{diam}(\mathcal{X})^2)$ and $C_2 = \sum_{t=1}^T \eta^{T-t}\alpha^{(t)}$.

*Proof.* The corollary is obtained by calculating $L_1^2$ and $L_l^2$ of Theorem 3. As the derivative of the proposed loss (i.e. Eq. 9) is bounded by 1, $L_l^2 \le 1$. $L_1$ can be derived as follows.

$$L_1^2 \overset{(a)}{\le} \sum_{i=1}^m \operatorname{lip}(h \leftarrow \boldsymbol{r}_i)^2 + \operatorname{lip}(h \leftarrow \boldsymbol{L})^2$$

$$\le 4m\operatorname{diam}(\mathcal{R})^2 + 4m^2\operatorname{diam}(\mathcal{R})^2\operatorname{diam}(\mathcal{X})^2,$$

where

$$\mathrm{lip}(h \leftarrow \boldsymbol{r})^2 \leq \left\| \frac{\partial h(\boldsymbol{x};\Theta)}{\partial \boldsymbol{r}} \right\|_2^2$$

$$\overset{(b)}{\leq} (2\,\mathrm{diam}(\mathcal{R}))^2,$$

$$\mathrm{lip}(h \leftarrow \boldsymbol{L})^2 \leq \left\| \frac{\partial h(\boldsymbol{x};\Theta)}{\partial \boldsymbol{L}} \right\|_F^2$$

$$= \left\| \sum_{i=1}^m 2(\boldsymbol{L}\boldsymbol{x} - \boldsymbol{r}_i)\boldsymbol{x}^T \exp(-\|\boldsymbol{L}\boldsymbol{x} - \boldsymbol{r}_i\|^2)\,\mathrm{sign}(\boldsymbol{r}_i) \right\|_F^2$$

$$= \mathrm{trace}\Big( \sum_{j=1}^m \sum_{i=1}^m 4\,\mathrm{sign}(\boldsymbol{r}_i)\,\mathrm{sign}(\boldsymbol{r}_j) \exp(-\|\boldsymbol{L}\boldsymbol{x} - \boldsymbol{r}_i\|^2)\cdot$$

$$\exp(-\|\boldsymbol{L}\boldsymbol{x} - \boldsymbol{r}_j\|^2)(\boldsymbol{L}\boldsymbol{x} - \boldsymbol{r}_i)\boldsymbol{x}^T\boldsymbol{x}(\boldsymbol{L}\boldsymbol{x} - \boldsymbol{r}_j)^T \Big)$$

$$\leq 4 \sum_{j=1}^m \sum_{i=1}^m \mathrm{trace}\Big( (\boldsymbol{L}\boldsymbol{x} - \boldsymbol{r}_j)^T(\boldsymbol{L}\boldsymbol{x} - \boldsymbol{r}_i)\boldsymbol{x}^T\boldsymbol{x} \Big)$$

$$\overset{(b)}{\leq} 4m^2\,\mathrm{diam}(\mathcal{R})^2\,\mathrm{diam}(\mathcal{X})^2,$$

and $\mathrm{sign}(\boldsymbol{r}_i) = 1$ if $\boldsymbol{r}_i$ is a representative instance of the positive class and $-1$ otherwise.

Inequality (a) is due to $\mathrm{lip}(h \leftarrow \boldsymbol{\theta}) \leq \sqrt{\sum_{q=1}^Q \mathrm{lip}(h \leftarrow \boldsymbol{\theta}_{[q]})}$ [3]; inequality (b) makes use of the assumption that $\{\boldsymbol{L}\boldsymbol{x}\} \subseteq \mathcal{R}$. $\qquad\square$

### C.5 Updating equations of SMILE

The updating equations of SMILE are as follows:

$$\boldsymbol{r}_j^{+,t+1} = \boldsymbol{r}_j^{+,t} - 2\lambda\alpha \boldsymbol{r}_j^{+,t} + \frac{\alpha}{n} \sum_{i=1}^n y_i l'(y_i h(\boldsymbol{x}_i;\Theta)) \exp(-\|\boldsymbol{L}\boldsymbol{x}_i - \boldsymbol{r}_j^+\|^2)(2\boldsymbol{r}_j^+ - 2\boldsymbol{L}\boldsymbol{x}_i)|_{\Theta^t};$$

$$\boldsymbol{r}_k^{-,t+1} = \boldsymbol{r}_k^{-,t} - 2\lambda\alpha \boldsymbol{r}_k^{-,t} - \frac{\alpha}{n} \sum_{i=1}^n y_i l'(y_i h(\boldsymbol{x}_i;\Theta)) \exp(-\|\boldsymbol{L}\boldsymbol{x}_i - \boldsymbol{r}_k^-\|^2)(2\boldsymbol{r}_k^- - 2\boldsymbol{L}\boldsymbol{x}_i)|_{\Theta^t};$$

$$\boldsymbol{L}^{t+1} = \boldsymbol{L}^t - 2\lambda\alpha \boldsymbol{L}^t + \frac{\alpha}{n} \sum_{i=1}^n y_i l'(y_i h(\boldsymbol{x}_i;\Theta)) \sum_j \exp(-\|\boldsymbol{L}\boldsymbol{x}_i - \boldsymbol{r}_j^+\|^2)2(\boldsymbol{L}\boldsymbol{x}_i - \boldsymbol{r}_j^+)\boldsymbol{x}_i^T|_{\Theta^t}$$

$$- \frac{\alpha}{n} \sum_{i=1}^n y_i l'(y_i h(\boldsymbol{x}_i;\Theta)) \sum_k \exp(-\|\boldsymbol{L}\boldsymbol{x}_i - \boldsymbol{r}_k^-\|^2)2(\boldsymbol{L}\boldsymbol{x}_i - \boldsymbol{r}_k^-)\boldsymbol{x}_i^T|_{\Theta^t}.$$

## D  Data description

Table 1 lists information on sample size and feature dimension, as well as the source of studied datasets.

## Footnotes

[1] In the cases of $\boldsymbol{m}$ being a matrix, the matrix will be reshaped into a vector and the vector $L_2$-norm can then be used; this is equivalent to using the matrix Frobenius norm directly.

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

Table 1: Data description

| Dataset | Source | #Instances | #Features |
|---|---|---|---|
| Australian | UCI | 690 | 14 |
| Cancer | UCI | 699 | 9 |
| Climate | UCI | 540 | 18 |
| Credit | UCI | 653 | 15 |
| German | UCI | 1000 | 24 |
| Haberman | UCI | 306 | 3 |
| Heart | UCI | 270 | 13 |
| ILPD | UCI | 583 | 10 |
| Liver | UCI | 345 | 6 |
| Pima | UCI | 768 | 8 |
| Ringnorm | Delve | 7400 | 20 |
| Twonorm | Delve | 7400 | 20 |

[4] John H Hubbard and Barbara Burke Hubbard. *Vector calculus, linear algebra, and differential forms: a unified approach*. Matrix Editions, 2015.

[5] H Quang Minh and Thomas Hofmann. Learning over compact metric spaces. In *COLT*, pages 239–254. Springer, 2004.

[6] Mehryar Mohri, Afshin Rostamizadeh, and Ameet Talwalkar. *Foundations of Machine Learning*. MIT Press, 2012.

[7] N Srebro and K Sridharan. Note on refined Dudley integral covering number bound, 2010.

[8] M. J. Wainwright. *High-dimensional Statistics: A Non-asymptotic Viewpoint*. Cambridge University Press, 2019.

[9] Nik Weaver. *Lipschitz Algebras*. World Scientific, 1999.