[Reviews · NeurIPS 2020]

Review 1

Summary and Contributions: This paper investigates the problem of deriving a generalization bound for models learned through a gradient descent approach, even when the objective function is non-convex. The main findings derived from the bound are that (i) early stopping is indeed beneficial for generalization purpose, (ii) having fewer parameters leads to models that generalize better, (iii) using classifiers and losses with small Lipschitz smooth constants is beneficial. From an algorithmic point of view, a new Metric Learning method called SMILE and based on these insights is proposed and its empirical performances is demonstrated on several datasets.

Strengths: - The paper has theoretical, algorithmic, and empirical contributions. - From a theoretical point of view, the proposed bound seems to be new from a Metric Learning point of view. Furthermore, it is general enough to be of potential interest for a broader range of researcher. - The proposed algorithm is simple yet intuitive. Furthermore, the theoretical insights provided by the bound are adequately used (smooth loss and classifiers, early stopping scheme, enforced small Lipschitz constants). - The empirical evaluation is quite convincing. It compares SMILE to 9 baselines on 12 datasets. Furthermore, the paper includes a small experiment demonstrating the interest of early stopping and the consequences on the generalization bound of using too many gradient steps.

Weaknesses: - Some of the theoretical findings are not so surprising (early stopping, fewer parameters). - The link between SMILE and the theoretical bound could be stronger in the sense that both of them could exist independently of each other. In particular, the exact bound corresponding to SMILE is never stated (given the exact Lipschitz constants nor the number of parameters for example).

Correctness: The claims, method, and empirical methodology appear to be correct.

Clarity: The paper is well written and easy to follow. Nevertheless, there is a few points that could be improved and a few typos: - Lines 104-112: The first part of the paragraph is a bit confusing at a first glance. Since only parametric classifiers are considered in the remainder of the paper, it would be better to focus on this setting only. Same thing in the following paragraph. - Lines 168-173: Instead of (i) and (ii), it would be better to directly refer to the previous equations (3) and (4). - Line 195: It seems to me that G is a function of the model but also of the training set and the index i. While this dependence disappears when computing the Lipschitz constant, it is a bit confusing at a first glance. - Line 223: The term \alpha has not been defined. There is also a few typos in the supplementary: - Line 81, Equation (a): z^n \in should be added. - Line 87: m^G_\mathcal{H} is used while it seems to be n^G_\mathcal{H} in the main paper. - Line 92: The last w should be a \hat{w}. - Line 109: This sentence is confusing and contains a false equality statement, it should be removed altogether. Furthermore, as mentioned before G should be a function of the model, the training, and the index i.

Relation to Prior Work: The connection between this work and previous work seems to be adequately discussed.

Reproducibility: Yes

Additional Feedback: 1) The first inequality below Line 67 in the supplementary is only true if the loss considered is positive. This should be stated as an assumption on the loss in the main paper. 2) In its current form, the full training set for each gradient steps. However, this is impractical when the number of examples increases. It would be interesting to study whether the proposed approach can be extended to batch gradient descent where only a subset of the data is considered at each iteration. 3) It would have been interesting to state the exact form of the theoretical bound for SMILE. Furthermore, it would have been interesting to study the influence of the parameter \lambda (introduced in Equation (11)) on the empirical performance of the algorithm and the tightness of the bound. --- The rebuttal only partially addressed my comments. The rebuttal provided experiments using batch gradient descent but did not formally show the impact of such an optimization scheme on the generalization bound (comment 2). It also provided experiments studying the influence of \lambda. However, the exact generalization bound corresponding to SMILE is still missing (comment 3). On the other hand, reading the other reviews (mainly R2), I realize that I have slightly overestimated the novelty of the technical contribution (proof technique, algorithm).


Review 2

Summary and Contributions: The paper looks at the problem of Mahalanobis metric learning and its application to binary classification. The main challenge addressed in this paper is a novel way of establishing a train time-excess risk tradeoff by choosing to establish a generalization PAC bound instead of a uniform convergence bound. The generalization PAC bound is then used to nevertheless enable agnostic learning. The generalization PAC bound is established by using a trick that exploits "local" uniform convergence in the parameter space rather than global uniform convergence that is harder to establish, especially when working with early stopping and other operations that limit search in the parameter space. These lead to a bound that can be seen as favoring early stopping. The paper also proposes a metric learning algorithm that learns a few prototypes per class and a Mahalanobis metric using a Huberized version of the hinge loss function. The overall classifier simply compares the average Gaussian kernel value (using the Mahalanobis distance in place of the Euclidean distance) to the prototypes of each class to perform classification.

Strengths: Soundness of the claims (theoretical grounding, empirical evaluation) -------------------------------- The theoretical guarantees are explained well and in detail. Significance and novelty of the contribution -------------------------------- The generalization PAC bound decomposition used to establish a high confidence generalization gap is interesting although the overall technique of establishing generalization PAC guarantees instead of uniform convergence guarantees has been explored before. Relevance to the NeurIPS community -------------------------------- The theoretical techniques may be of interest in establishing similar time-generalization tradeoff bounds for non-convex problems where the traditional route of uniform convergence does not make much sense.

Weaknesses: Soundness of the claims (theoretical grounding, empirical evaluation) -------------------------------- The paper establishes an interesting theoretical guarantee but the SMILE algorithm does not seem to exploit it very much. The SMILE algorithm is basically the Nadaraya-Watson estimator (with the Gaussian kernel with the Mahalanobis metric instead of the Euclidean metric) where the support vectors are also learnt instead of using training points as support vectors. It is not clear how much advantage does the SMILE classifier get by learning the representative instances. I suspect that they may account for a lot of the advantage SMILE has over other algorithms, since the SMILE classifier is the very simple Nadaraya-Watson estimator, as pointed out above. Moreover, the other competitor algorithms were not offered a chance to similarly learn nice prototypes. One way to rebut this criticism would be to run SMILE but restrict it to using a subset of training points or else award all other methods e.g. LMNN the luxury of choosing representative points. Significance and novelty of the contribution -------------------------------- Lemma 1 is the starting point of all generalization error bounds that use uniform convergence and is not novel. Relevance to the NeurIPS community -------------------------------- Metric learning by itself is not the state-of-the-art in most areas of machine learning. It would have been nice if some contemporary applications with VAE or deep metric learning could have been explored.

Correctness: The theoretical bounds seem correct. Empirical work is satisfactory in terms of choice of a large number of competitors and the experiment that seeks to validate the concentration phenomenon suggested by the theoretical bounds (Figure 2). However, datasets are very small and the experimental setup is a bit outdated in terms of using plain Mahalanobis metric learning to perform classification.

Clarity: Yes, the paper is easy to read.

Relation to Prior Work: Yes, relation to prior work is described very well.

Reproducibility: Yes

Additional Feedback: Please give the definition of agnostic PAC learnability in the paper. This definition is crucial since the whole point in the theoretical analysis is to avoid uniform convergence. Since the paper overcomes a common hurdle of requiring convex formulations or even optimization accuracy, it is a shame that the paper holds back and does not explore representation learning fully e.g. using deep networks. This is important since metric learning is a classical form of representation learning, deep learning being a more contemporary one.


Review 3

Summary and Contributions: This paper presents a new route to the generalization guarantee on classifiers optimized via GD, considering the influence of sampling randomness to the concentration property of parameters and embracing algorithmic parameters. The authors propose a new decomposition theorem to obtain the generalization PAC bound, which consequently guarantees the agnostic PAC learnability.

Strengths: The theoretical analysis of the paper is completed. This paper is related with the NeurIPS community. The empirical evaluation of this paper is ok. In addition, this paper presents a new route to the generalization guarantee on classifiers optimized via GD, considering the influence of sampling randomness to the concentration property of parameters and embracing algorithmic parameters. The authors propose a new decomposition theorem to obtain the generalization PAC bound, which consequently guarantees the agnostic PAC learnability. They demonstrate the necessity of Lipschitz smooth classifiers and loss functions for generalization and theoretically justify the benefit of early stopping.

Weaknesses: However, there are some limitations in this paper: 1. The motivation of this paper is not clear. The authors did provide enough proofs to show the importance of the proposed results. The authors have been showing their theoretical derivation, but has not explained the significance of your proved results. 2. Why do not the authors compare the proposed results with other related methods? For example, compared with generalization bounds of metric learning [R1], what are the advantages of your results? I cannot sure if the given results of this paper is meaningful since the authors did not provide any explanations. 3. Which aspect of your approach determines the performance of the model (14)?Loss term or regularization term? You should provide corresponding analysis. Overall, even though I read this paper for some times, I didn't get any meaningful and interesting knowledge from this paper. Meanwhile, I am not sure if this proposed method can be applied to other fields. [R1] Cao Q, Guo Z C, Ying Y. Generalization bounds for metric and similarity learning[J]. Machine Learning, 2016, 102(1): 115-132. %%%%%%%%%%%%%%%% The authors' response is convincing. Thus, I consider increasing my scores.

Correctness: I have checked the main steps of learning frameworks and optimization. Some main claims are not satisfied. But he empirical analysis is correct.

Clarity: The paper is well organized and written. However, some issues as in weakness should be carefully considered to further improve the paper.

Relation to Prior Work: No. I think the authors need to further discuss this problem.

Reproducibility: Yes

Additional Feedback:


Review 4

Summary and Contributions: The manuscript mainly presents some theoretical analysis for metric learning algorithms with non-convex objective functions. Based on the theoretical results, a novel metric learning algorithm is developed.

Strengths: 1. The studied problem of (theory of non-convex metric learning) is important. 2. The proposed theoretical results seem reasonable and novel. 3. A novel metric learning algorithm is also developed based on the theoretical analysis.

Weaknesses: 1. The proposed method is only applicable to binary classification problems, and thus the application is limited; 2. There are some inappropriate statements, e.g., it is not appropriate to claim that “training time” is an algorithmic parameter. 3. The proposed method is designed for non-convex metric learning, and thus some recently proposed deep metric learning algorithms should be included for comparison.

Correctness: Seems correct but not check very carefully.

Clarity: Should be improved.

Relation to Prior Work: Clear enough.

Reproducibility: Yes

Additional Feedback: NA.

[Author Response · NeurIPS 2020]

We thank all the reviewers (**R1**-**R4**) for their insightful comments and invaluable feedback on our manuscript. As
the reviewers mentioned, our work shows the following strengths. (1) The proposed decomposition theorem and the
resulting bound are interesting (R1,R2) and new (R3,R4); (2) The theoretical study of non-convex metric learning is
important (R4), and the technique is general and of interest in establishing similar time-generalization tradeoff (R1,R3);
(3) Our algorithm is novel (R4), intuitive, and adequately uses the theoretical insights provided by the bound (R1);
(4) The theoretical analysis is complete (R2,R3), and the paper is easy to read (R1,R2,R3). We respond to the key
comments below but will address all feedback in the final version.

**[R3] Motivation.** Our motivation is to provide an approach to deriving a new generalization bound that considers the
parameters related to optimization, such as the number of iterations, which cannot be achieved by current uniform
convergence bounds. Then as summarized by **R1**,"the theoretical insights provided by the bound are adequately used
(smooth loss and classifiers, early stopping scheme, enforced small Lipschitz constants)" in the proposed algorithm.

**[R3] Significance of the proved results.** As pointed out by **R1**, "the proposed bound seems to be new from a metric
learning point of view. Furthermore, it is general enough to be of potential interest for a broader range of researcher".
As suggested by **R2**, "the theoretical techniques may be of interest in establishing similar time generalization trade-off
bounds for non-convex problems, where the traditional route of uniform convergence does not make much sense" and
"the paper overcomes a common hurdle of requiring convex formulation or even optimization accuracy". In Sec.3.3.4,
we also provide discussion on the implications of Theorem 3. Greater stress on the significance of the proved results
and their wider applications will be added to Sec.3.3.4, Abstract and Introduction to make the significance more clear.

**[R3] Compared with generalization bounds of Cao et al. [6].** The bounds obtained in [6] is based on the framework
of uniform convergence. However, as pointed out by **R2**, when considering "time-generalization trade-off bounds", i.e.,
the generalization bounds considering the iteration number, "the traditional route of uniform convergence does not make
much sense". The approach proposed by our paper is fundamentally different from uniform convergence and it takes
optimization parameters into account. We will, on top of discussing the shortcomings of uniform convergence, highlight
the difference between our approach and their approaches including [6] in Introduction to make this more clear.

**[R1] Study the influence of the parameter $\lambda$; [R3] Performance of the model determined by loss term or regu-**
**larization term?** Figs.(a) and (b) show the influence of $\lambda$ on the generalization gap, training accuracy and test accuracy.
The generalization gap decreases with $\lambda$, which is consistent with our theoretical result that constraining the norms of
$L, x, r$ gives smaller Lipschitz constants, thereby tightening the bound. As the training accuracy generally decreases
with $\lambda$ as well, the test accuracy is highest when $\lambda = 0.5$. More detailed analysis will be added into the final version.

(a) Effect of $\lambda$ on the generalization gap.     (b) Effect of $\lambda$ on training and test accuracy.     (c) Concentration of deep ML.

**[R2, R4] Explore with deep metric learning (ML) algorithms; [R1] Extension to batch gradient descent.** We
have investigated the impact of training iterations on the concentration of parameters of deep ML and present the result
in Fig.(c). The network consists of three convolutional blocks and one fully connected (FC) layer and is trained with
mini-batch gradient descent (batch size=100); $\epsilon_{100\%}$ is calculated from the parameters of the FC layer over 10 rounds.
Similar to Fig.2 in the main paper, we see again that $\epsilon_{100\%}$ increases along training, indicating that the variance of
learned parameters becomes larger. While the empirical result is encouraging, we acknowledge that the proved theorem
cannot guarantee the learnability of mini-batch/stochastic gradient descent algorithms due to the randomness of training
instances introduced in each batch. This is indeed important and will be added into the Broader Impact Section.

**[R4] The method is only applicable to binary classification problems**. Our theoretical framework can be readily
generalized to multi-class classification. Theorem 2 and Lemma 2 are proved regardless of the number of classes.
Lemma 3 needs to be revised by considering the uniform convergence property of the hypothesis class defined over a
multi-class risk $R$. This has been discussed in some references. For example, based on the risk defined on p185 of
[Ref 1], uniform convergence is guaranteed by Theorem 8.1 on p187. More details will be added into the final version.
[Ref 1] Mohri, M., Rostamizadeh, A., & Talwalkar, A. (2018). Foundations of Machine Learning.

**[R2] Advantage gained by learning the representative instances.** We will add experiments by awarding other
methods, such as LMNN, the luxury of choosing representative points, as suggested by **R2**.

[Meta-Review · NeurIPS 2020]

The four referees support acceptance for the contribution and I also recommend acceptance. However, please consider revising your paper in order to address the lack of novelty of Lemma 1, to add the generalization bound of SMILE and to precise the fact that your result is rather general and not purely specific to metric learning.